# P(*all-atom*) Is Unlocking New Path For Protein Design

## Abstract

We introduce Pallatom, an innovative protein generation model capable of producing protein structures with all-atom coordinates. Pallatom directly learns and models the joint distribution $P(structure, seq)$ by focusing on $P(all\text{-}atom)$, effectively addressing the interdependence between sequence and structure in protein generation. To achieve this, we propose a novel network architecture specifically designed for all-atom protein generation. Our model employs a dual-track framework that tokenizes proteins into token-level and atomic-level representations, integrating them through a multi-layer decoding process with "traversing" representations and recycling mechanism. We also introduce the `atom14` representation method, which unifies the description of unknown side-chain coordinates, ensuring high fidelity between the generated all-atom conformation and its physical structure. Experimental results demonstrate that Pallatom excels in key metrics of protein design, including designability, diversity, and novelty, showing significant improvements across the board. Our model not only enhances the accuracy of protein generation but also exhibits excellent training efficiency, paving the way for future applications in larger and more complex systems.

## 1 Introduction

The theoretical foundation of protein modeling has been built upon two key conditional probability distributions: $P(structure \mid seq)$ and $P(seq \mid backbone)$. The former, $P(structure \mid seq)$, corresponds to the all-atom protein structure prediction task, which involves determining the three-dimensional structure of a protein given its amino acid sequence (Abramson et al., 2024; Jumper et al., 2021; Lin et al., 2023; Baek et al., 2023). The latter, $P(seq \mid backbone)$, underpins the fixed-backbone design task, where the goal is to identify a sequence that will fold into a given protein backbone structure (Dauparas et al., 2022; Hsu et al., 2022). In summary, these probability distributions has successfully advanced the field of protein engineering.

With the advancement of deep learning in protein science, two distinct approaches for protein design have emerged. One approach is the protein hallucination (Anishchenko et al., 2021), which explores the landscape of a $P(structure \mid seq)$ model using Monte Carlo or gradient-based optimization techniques. This method yields valid protein structures, but requires an additional $P(seq)$ model, such as protein language models (Rives et al., 2021), to correct or redesign the sequence. Essentially, this approach can be viewed as optimization process of $P(structure \mid seq) \cdot P(seq)$. Another approach attempts to explore the $P(backbone)$ distribution. a series of protein generation models based on SE(3) invariance or equivariance networks (Jing et al., 2020; Satorras et al., 2021) have recently emerged, these method rely on an additional $P(seq \mid backbone)$ process to determine the protein sequence. This optimization strategy can be regarded as $P(backbone) \cdot P(seq \mid backbone)$.

This step-wise design process has limitation in approximating the joint distribution through marginal distributions. The $P(structure \mid seq) \cdot P(seq)$ strategy faces challenges when sampling in the high-dimensional

sequence space, while the $P(backbone) \cdot P(seq \mid backbone)$ strategy fails to account for explicit side-chain interactions and is bottlenecked by the capability of the fixed-backbone design model.

The ultimate goal of protein generation is to directly obtain a sequence along with its corresponding structure, i.e., to develop a model capable of describing the joint distribution $P(structure, seq)$ or $P(backbone, seq)$. Recently, some studies have started to adopt co-generation approaches, such as model based on co-diffusion (Campbell et al., 2024) or co-design (Ren et al., 2024). While these methods primarily rely on SE(3) networks, they still separately model the backbone and sequence, without considering side-chain conformations and leading to an insufficient description of the structure. Protpardelle (Chu et al., 2024), an all-atom protein diffusion model, similarly adopts co-generation approaches with an explicit all-atom representation, taking a step further in the field. However, the experimental results indicate that the generated sequence fails to accurately encode the intended fold, necessitating an additional round of sequence redesign and side-chain refinement.

In this study, we introduce a novel approach for all-atom protein generation called **Pallatom**. Our extensive experiments show that by learning $P(all\text{-}atom)$, high-quality all-atom proteins can be successfully generated, eliminating the need to learn marginal probabilities separately. Inspired by AlphaFold3 (Abramson et al., 2024), we adopt a dual-track framework that tokenizes proteins into residues or atoms, and develop a novel module incorporating "traversing" representations and multi-layer decoding units. This module efficiently integrates and updates token-level and atomic-level representations through a dual-track recycling mechanism, enabling self-conditioned inferencing and enhancing information flow between blocks. Additionally, we propose a new amino acid coordinates representation, atom14, to address the challenge of representing unknown side-chain coordinates. Introducing virtual atoms to all amino acids type prevents sequence information leakage problem. The key insight of Pallatom is recognizing that all-atom coordinates of a protein inherently encode both structural and sequence information. Directly learning $P(all\text{-}atom)$ opens a new path for co-generative modeling of structure and sequence.

Our contributions are summarized as follows:

- We develop a network architecture for all-atom protein generation tasks, which effectively represents both protein backbones and sidechains.

- We explore the atom14 representation to achieve a unified description of unknown amino acid side-chain coordinates in generative tasks.

- We use our framework to develop Pallatom, a state-of-the-art all-atom protein generative model.

## 2 PRELIMINARIES

### 2.1 ALL-ATOM MODELING AND REPRESENTATION

The all-atom protein generation model faces many challenges in constructing both backbone and side-chain atoms. A pivotal initial question arises: "How to represent a system with a variable number of atoms?". At the initial sampling stage, both the backbone and sequence are unknown, however, the atom number of a system depends on unique sequence, once the sequence is determined, it also dictates the structure.

To avoid potential conflicts arising from the simultaneous design of sequence and structure, we define a representation called atom14, which pads the initial protein with $L$ residues as $\mathbf{x} = \{x^l\}_{l=1}^L \rightarrow \mathbf{x}_0$, considering it as a 3D point cloud distribution $P(\mathbf{x}_0) \in \mathbb{R}^{L \times 14 \times 3}$. For example, if residue $x^l$ is CYS, its coordinates $[\mathrm{N}, \mathrm{C}_\alpha, \mathrm{C}, \mathrm{O}, \mathrm{C}_\beta, \mathrm{S}_\gamma] \in \mathbb{R}^{6 \times 3}$ is padded with 8 virtual atoms that coincide with its $\mathrm{C}_\alpha$ position, resulting in $x^l \rightarrow x_0^l \in \mathbb{R}^{14 \times 3}$.

Our approach assumes that the all-atom distribution of the protein sidechains, including properties such as hydrophobicity, polarity, and even hydrogen bonds and salt bridges, are inherently encoded within the all-atom coordinates distribution. Even without knowing the specific element atom type, the conformationally similar amino acids, such as CYS and SER, can be distinguished from atomic-level features. Therefore, we additionally trained a single "visualization" head to predict the corresponding amino acid type. With discarding the redundant virtual atoms based on the predicted amino acid type as a post-processing step, we can generate all-atom proteins with corresponding sequences from a 3D point cloud. Additionally, we provide the alanine reference conformer features to guide the network in forming a stable backbone frame conformation.

## 2.2 DIFFUSION MODELING ON ALL-ATOM PROTEIN

The use of all-atom representation eliminates the constraints imposed by the complex forms of the SE(3) frame (Yim et al., 2023b) and the Riemannian diffusion framework (De Bortoli et al., 2022). Diffusion-based generative models using Gaussian noise distribution have a strong theoretical foundation, with various adaptations influenced by factors such as the sampling schedule, training dynamics (Song et al., 2021). We adopt the EDM framework (Karras et al., 2022) with slight modifications and employed a Gaussian diffusion model on $\mathbb{R}^{L \times 14 \times 3}$.

Assuming the data distribution of protein coordinates under the `atom14` representation as $p_{data}(\mathbf{x})$ with standard deviation $\sigma_{data}$, the forward process involves adding Gaussian noise of varying scales to generate a series of noised distributions $p(\mathbf{x}; \sigma) = \mathcal{N}(\mathbf{x}, \sigma^2 \mathbf{I})$. When $\sigma_{max} \gg \sigma_{data}$, $p(\mathbf{x}; \sigma_{max})$ approximates pure Gaussian noise. For a noise schedule $\sigma(t) = t$ (following EDM notation where $\sigma(t)$ indicates the noise schedule and $\sigma_t$ represents the noise level sampled from $p_{train}(t)$ at time $t$), the probability flow ordinary differential equation (ODE) is given by:

$$\mathrm{d}\mathbf{x} = -\sigma(t)\nabla_{\mathbf{x}} \log p(\mathbf{x}; \sigma(t))\mathrm{d}t \tag{1}$$

Here, $\nabla_{\mathbf{x}} \log p(\mathbf{x}; \sigma)$ is the score function, which does not depend on the normalization constant of the underlying density function $p(\mathbf{x}; \sigma)$. A neural network $D_\theta(\mathbf{x}, \sigma)$ is typically trained for each $\sigma_t$ using the following loss function to match the score function (Song et al., 2021):

$$\mathbb{E}_{\mathbf{x}_0 \sim p_{data}, \mathbf{x}_t \sim p(\mathbf{x}_0; \sigma_t)}[\lambda(\sigma)||D_\theta(\mathbf{x}_t, \sigma_t) - \mathbf{x}_0||_2^2], \quad \nabla_{\mathbf{x}} \log p(\mathbf{x}; \sigma) = (D_\theta(\mathbf{x}_t, \sigma_t) - \mathbf{x}_t)/\sigma_t^2 \tag{2}$$

Practically, we employed the EDM preconditioning technique, which resulted in improved generation performance. Consequently, we can derive $D_\theta$ and the loss function as:

$$D_\theta = c_{skip}(\sigma)\mathbf{x}_t + c_{out}(\sigma)F_\theta(c_{in}(\sigma)\mathbf{x_t}, c_{noise}(\sigma)) \tag{3}$$

$$\mathbb{E}_{\mathbf{x}_0, \mathbf{x}_t}[\lambda(\sigma)||c_{out}(\sigma)F_\theta(c_{in}(\sigma) \cdot \mathbf{x}_t, c_{noise}(\sigma)) - (\mathbf{x}_0 - c_{skip}(\sigma) \cdot \mathbf{x}_t)||_2^2] \tag{4}$$

where $c_{skip}(\sigma) = \sigma_{data}^2/(\sigma^2 + \sigma_{data}^2)$, $c_{out}(\sigma) = \sigma \cdot \sigma_{data}/\sqrt{\sigma_{data}^2 + \sigma^2}$, $c_{in}(\sigma) = 1/\sqrt{\sigma^2 + \sigma_{data}^2}$, $c_{noise}(\sigma) = \frac{1}{4}\ln(\sigma)$ represent skip scaling, output scaling, input scaling, and noise conditioning, respectively. We set $\lambda(\sigma) = 1/c_{out}(\sigma)^2$.

## 2.3 FRAMEWORK FOR ALL-ATOM PROTEIN GENERATION

AlphaFold3 provided an excellent initial framework for all-atom protein generation. However, the generation task fundamentally differs from the structure prediction task, requiring us to tailor the structure prediction framework with several modifications. In structure prediction, numerous co-evolutionary signals can typically be extracted from homologous sequences, which can then be decoded into a three-dimensional

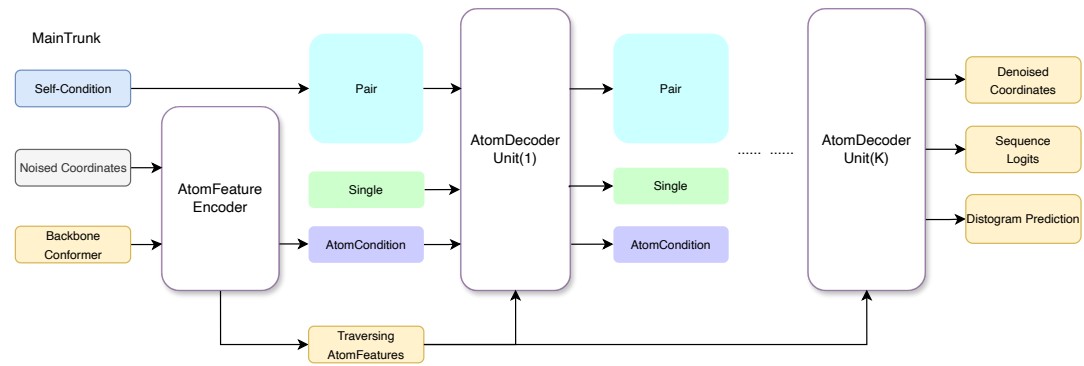

Figure 1: Pallatom model framework.

structure. In contrast, the generation task begins with noised coordinates, requiring the model to learn how to extract structural information iteratively.

Providing self-conditions (Chen et al., 2022) is an effective way to enhance generation performance. A simple approach is to integrate the information extraction modules directly into the structural decoder, allowing the partially denoised information from each block to be used as self-conditions for the subsequent block. More specifically, in our modifications, the denoised coordinates from each block are converted into a pairwise feature, which is reinjected into pair representation and used as self-conditions in the next block to update single representation. These modifications effectively address the self-conditioning pipeline between blocks.

In practice, the interaction and updating of dual-track protein representations within multiple decoder units present new challenges. We find that if residual connections are used simultaneously for both token-level and atomic-level representations across multiple decoder units, the token-level representation tends to be repeatedly broadcast and inappropriately accumulated. This not only leads to numerical instability in atomic-level representations but also disrupts the model's performance. Therefore, we propose using traversal atomic-level representations to carry the token-level representations within the current decoding unit. These two modifications enable the new framework to effectively generate all-atom protein structures. Details can be found at Figure 1 and 3.

## 3 METHOD

### 3.1 MAINTRUNK: THE DENOISNG NETWORK

We refer to the protein generation task as the generation of all-atom coordinates. In the `atom14` representation, a protein with $L$ residues can be expressed as $\mathbf{x}_0 = \{x_0^l\}_{l=1}^{L}$, which $x_0^l \in \mathbb{R}^{14 \times 3}$ represents the all-atom coordinates of a residue. For each time step $t$ in the diffusion process, the network predicts the updated coordinates from the input $\mathbf{x}_t \sim \mathcal{N}(\mathbf{x}_0, \sigma_t^2 \mathbf{I})$.

The network comprises two main components: an input encoding module and multiple iterative decoding units. Figure 1 illustrates the main architecture. We adopt the dual-track framework, the atomic-level representation for atoms using local attention mechanisms and the token-level representation for residues using global attention mechanisms.

For feature initialization and encoding, we utilize coordinates feature from the standard conformation of alanine, the positional encoding within the residues and the noisy coordinates vector to initialize 1D atomic-level representations. The 1D representations are then updated using a 3-layer AtomTransformer encoder, with the initial atomic pair representations as attention bias. These atomic-level representations were re-garded as static traversing atomic-level representations ($c_l^{\text{skip}}$, $q_l^{\text{skip}}$ and $p_{lm}^{\text{skip}}$ in Figure 3). The single repre-sentation initialization is performed using the diffusion timestep feature and the positional embedding. The pair representation is initialized using the relative positional embedding introduced in AlphaFold2 (Jumper et al., 2021) and a self-conditioned template distogram feature from the previous prediction. The interac-tion between token-level and atomic-level information is propagated through broadcasting and aggregation. Detailed features are recorded in Appendix Table 3.

For the decoding part, we employ an iterative update mechanism where token-level representations are broadcast to the atomic level, and atomic information is then fed back through a recycling process. To address the issue of residual connections in the atomic representation mentioned earlier, for each block, we use an intermediate atomic-level representations, which is composed of broadcast token-level representations and traversing atomic-level representations, to predict coordinates updates $\mathbf{x}_{update}^{(k)}$, the current predicted structure $\mathbf{x}_0^{(k)}$ using the cumulative updates from the first $k$ units: $\mathbf{x}_0^{(k)} = c_{skip}(\sigma_t) \cdot \mathbf{x}_t + c_{out}(\sigma_t) \cdot \mathbf{x}_{updates}^{(k)}$.

After the coordinates are denoised, they are transformed to relative distance matrix $\mathbf{x}_0^{(k)}$ and recycled back into the pair representation via triangle attention layer, effectively addressing the challenges of updating pair representations. In practice, we found that recycling pair information accelerates model training and en-hances inference capabilities. Utilizing the minimal decoding unit and supervised training on intermediate coordinates simplifies the scaling of the network and allows for increasing its depth. Through the iterative 8-layer decoder, we effectively updated the atomic-level and token-level representations. The cumulative updating mechanism for coordinates prediction allows the model to gradually refine these predictions, ul-timately leading to the realistic all-atom protein structure. Details can be found at Appendix Algorithm 2.

## 3.2  SEQHEAD: SEQUENCE DECODER

To convert the generated coordinates into a real protein, we need a module that translates the position infor-mation into an amino acid sequence. We add a `SeqHead` to each decoding unit. Specifically, we aggregate the updated 1D atomic-level representations from the AtomAttentionDecoder corresponding to each token and then employ a linear layer to predict the logits for the 20 amino acid types $\hat{\mathbf{a}}^{(k)} \in \mathbb{R}^{L \times 20}$. We take the predictions from the last unit as the final sequence logits, $\hat{\mathbf{a}} = \hat{\mathbf{a}}^{(K)}$.

## 3.3  TRAINING LOSS

Our training method mainly follows the application and improvements of the EDM framework. The de-noising all-atom positions score-matching losses are described according to Eq.(2). Given that the network architecture lacks inherent equivariance constraints and iteratively refines coordinates across multiple decod-ing stages, it is imperative that the coordinates transformations between these stages remain invariant under changes in orientation to ensure consistent geometric interpretations. Therefore, we employ an aligned MSE loss similar to that in Alphafold3. We first perform a rigid alignment of the ground truth $\mathbf{x}_0$ on the denoised structure $\hat{\mathbf{x}}_0$ as $\mathbf{x}_0^{\text{aligned}}$. The MSE loss is then defined as:

$$\mathcal{L}_{\text{atom}} = \frac{||\hat{\mathbf{x}}_0 - \mathbf{x}_0^{\text{aligned}}||^2}{3L}$$

For sequence decoding, we use the standard cross-entropy loss function to evaluate the difference between the predicted sequence $\hat{\mathbf{a}}$ and the true sequence $\mathbf{a}_0$. The loss function is defined as $\mathcal{L}_{\mathrm{seq}} = \mathrm{CE}(\hat{\mathbf{a}}, \mathbf{a}_0)$. The basic loss function of the network is:

$$\mathcal{L}_0 = \lambda(\sigma_t) \cdot \mathcal{L}_{\mathrm{atom}} + \alpha_0 \cdot \mathcal{L}_{\mathrm{seq}}$$

To capture the fine-grained characteristics of the all-atom structure, we introduce the smooth local distance difference test (LDDT) loss $\mathcal{L}_{\mathrm{smooth\_lddt}}(\hat{\mathbf{x}}_0, \mathbf{x}_0)$ from AlphaFold3. The specific algorithm can be found in the Appendix Algorithm 8. To supervise the updating of pair-wise features, we employ the distogram head loss to constrain the global distance distribution at the token level. The pair representation is symmetrized and projected into 64 distance bins with probability $p_{ij}^b$, and supervised with one-hot encoded target bins $y_{ij}^b$. Similarly, to supervise the local relative distance distribution at the atomic-level, we project the 2D features from local atomic attention into 22 distance bins from 0 $\mathring{A}$ to 10 $\mathring{A}$ with $q_{nm}^b$, and construct the loss with one-hot encoded target bins at the atomic-level. Here, the local region corresponds to the area calculated within the local attention on the $14L \times 14L$ atomic-level map.

$$\mathcal{L}_{\mathrm{dist\_token}} = -\frac{1}{L^2} \sum_{i,j} \sum_{b=1}^{64} y_{ij}^b \log p_{ij}^b, \quad \mathcal{L}_{\mathrm{dist\_atom}} = -\frac{1}{NM} \sum_{n,m \in \mathrm{local}} \sum_{b=1}^{22} y_{nm}^b \log q_{nm}^b$$

In the early stages of development, we discovered that supervising the intermediate sequences and structures predicted by each decoder unit significantly enhanced both network performance and inference stability. Furthermore, we introduced a loss weight decay mechanism with $\gamma = 0.99$ between different blocks, assigning greater weight to the later layers.

$$\mathcal{L}_{\mathrm{med}}^k = \lambda(\sigma_t) \cdot \frac{||\mathbf{x}_0^{(k)} - \mathbf{x}_0^{\mathrm{aligned}}||^2}{3L} + \alpha_0 \cdot \mathrm{CE}(\mathbf{a}_0^{(k)}, \mathbf{a}_0), \quad \mathcal{L}_{\mathrm{med}} = \frac{1}{K} \sum_{k=1}^{K} \gamma^{K-k} \cdot \mathcal{L}_{\mathrm{med}}^k$$

The total loss can be written as:

$$\mathcal{L} = \mathcal{L}_0 + \alpha_1 \cdot \mathcal{L}_{\mathrm{smooth\_lddt}} + \alpha_2 \cdot \mathcal{L}_{\mathrm{dist\_token}} + \alpha_3 \cdot \mathcal{L}_{\mathrm{dist\_atom}} + \alpha_4 \cdot \mathcal{L}_{\mathrm{med}}$$

### 3.4 SAMPLING

The sampling process is described in Algorithm 1. The modules highlighted in blue are identical to those proposed in AlphaFold3. The initial atoms are sampled from a Gaussian distribution on $\mathbb{R}^{L \times 14 \times 3}$. We only use the first-order Euler method as the ODE solver with $T$ steps for discretization. Optionally, additional noise can be injected during the sampling steps to introduce stochasticity into the ODE solving process. We focus only on the sequence distribution decoded by the network in the final sampling step, employing a low-temperature softmax strategy to derive an approximate discrete one-hot amino acid sequence as the final sequence.

## 4 EXPERIMENTS

### 4.1 TRAINING SETTING

The training dataset of the model includes the PDB (Zardecki et al., 2022) and AlphaFold Database (AFDB) (Varadi et al., 2021). We performed rigorous data cleaning on augmented data from AFDB to obtain high-quality results. Detailed descriptions can be found in the appendix. We focus on small monomer proteins that can be easily synthesized using commercial oligo-pool method and the models are trained on crops of lengths up to 128. The model training utilized the Adam optimizer Kingma & Ba (2017) with a learning rate of 1e-3, $\beta_1 = 0.9$, $\beta_2 = 0.999$, and a batch size of 32. Details are provided in the Appendix.

---

**Algorithm 1** Pallatom Inference

---

1: **def SampleDiffusion** ($\{f^*\}, T = 200, \lambda = 1.003, \eta = 2.25, \gamma_0 = 0.2, t_{min} = 0.01, t_{max} = 1.0$):
2: $\delta_t = 1/T$
3: $c_T = \text{GetNoiseSchedule}(1 - \text{uniform}(0, 1) \cdot \delta_t)$
4: $\vec{\mathbf{r}}_l \sim c_T \cdot \mathcal{N}(\vec{0}, \mathbf{I}_3)$
5: **for all** $t \in \{T, \dots, 1\}$ **do**
6:     $t_p = t/T - \text{uniform}(0, 1) \cdot \delta_t$
7:     $c_\tau = \text{GetNoiseSchedule}(t_p)$
8:     $c_{\tau-1} = \text{GetNoiseSchedule}(t_p - \delta_t)$
9:     $\vec{\mathbf{r}}_l \leftarrow \text{\textcolor{blue}{CentreRandomAugmentation}}(\vec{\mathbf{r}}_l)$
10:     $\gamma = \gamma_0$ if $t_{min} \leq t/T \leq t_{max}$ else $0$
11:     $\hat{t} = c_\tau(\gamma + 1)$
12:     $\vec{\mathbf{r}}_l^{nosiy} = \vec{\mathbf{r}}_l + \lambda\sqrt{\hat{t}^2 - c_\tau^2} \cdot \mathcal{N}(\vec{0}, \mathbf{I}_3)$
13:     $\vec{\mathbf{r}}_l^{denoised}, \mathbf{f}_i^{\text{seq\_logits}} = \text{MainTrunk}(\{f^*\}, \vec{\mathbf{r}}_l^{nosiy}, \hat{t}, t_p)$
14:     $\vec{\delta}_l = (\vec{\mathbf{r}}_l^{nosiy} - \vec{\mathbf{r}}_l^{denoised})/\hat{t}$
15:     $dt = c_{\tau-1} - \hat{t}$
16:     $\vec{\mathbf{r}}_l \leftarrow \vec{\mathbf{r}}_l^{noisy} + \eta \cdot dt \cdot \vec{\delta}_l$
17: **end for**
18: **return** $\{\vec{\mathbf{r}}_l\}, \{\mathbf{f}_i^{\text{seq\_logits}}\}$

---

## 4.2 METRICS

While some evaluation criteria used for protein backbone generation are not suited for the new task, we propose new metrics specifically designed for assessing all-atom protein generation.

The first criterion is structure designability. The traditional self-consistency process assesses the designability of protein backbones (**DES-bb**). This involves using a fixed-backbone design model (e.g., ProteinMPNN (Dauparas et al., 2022)) to generate $N_{seq}$ sequences for the backbone, which are then folded by structure prediction models like ESMfold (Lin et al., 2023). The backbone's designability is evaluated by the optimal TM-score or $C_\alpha$-RMSD between the folded and original backbones. However, this metric is not suitable for evaluating all-atom proteins, which include side-chain atoms. Therefore, we similarly define the designability of all-atom protein generation, denoted as **DES-aa**. For the all-atom proteins, the sequence is used to predict the structure, and the sample is considered designable if the mean pLDDT of the predicted structure exceeds 80 and the all-atom RMSD (aaRMSD) is less than 2 Å. This metric ensures high atomic-level accuracy and provides strong confidence in the structural integrity and designability of the predicted protein, indicating that the sequence is likely to adopt its intended native fold.

The second criterion is structure diversity, denoted as **DIV-str**. This can be quantified by calculating the clusters number of the designable structures using Foldseek (Van Kempen et al., 2024). For all-atom proteins, we use a similar diversity evaluation method for the generated sequences, denoted as **DIV-seq**. Specifically, we use MMseq2 (Steinegger & Söding, 2017) to calculate the clusters number of the designable sequences.

The last criterion is structure novelty, which evaluates the structural similarity between the generated backbones and natural proteins in the PDB, denoted as **NOV-str**. This is calculated by the TM-score of the generated designable backbones compared to the most similar proteins in the PDB.

In our evaluation, we use the following two modes:

Table 1: Comparison of various methods. Protpardelle utilized ProteinMPNN as an auxiliary tool in all-atom proteins generation, resulting in identical results for CO-DESIGN 1 and PMPNN 1. In the case of Multiflow, which can only generate protein backbones and sequences without side chains, the reported DES-aa metric is based on $C_\alpha$RMSD rather than aaRMSD.

| Method | CO-DESIGN 1 | | | PMPNN 1 | | |
|---|---|---|---|---|---|---|
| | DES-aa (↑) | DIV-str/seq (↑) | NOV-str (↓) | DES-bb (w/wo) (↑) | DIV-str (↑) | NOV-str (↓) |
| Protpardelle* | 30.00% | 15 / 26 | 0.747 | 30.00% (80.23%) | 15 | 0.747 |
| ProteinGenerator | 43.14% | 83 / 706 | 0.791 | **93.14% (96.34%)** | 151 | 0.785 |
| Multiflow* | 62.74% | 134 / 1042 | 0.753 | 84.69% (95.43%) | 184 | 0.744 |
| RFdiffusion | N/A | N/A | N/A | 78.29% (84.40%) | 165 | 0.805 |
| Pallatom | **85.03%** | **291 / 1466** | **0.719** | 89.89% (94.46%) | **318** | **0.716** |

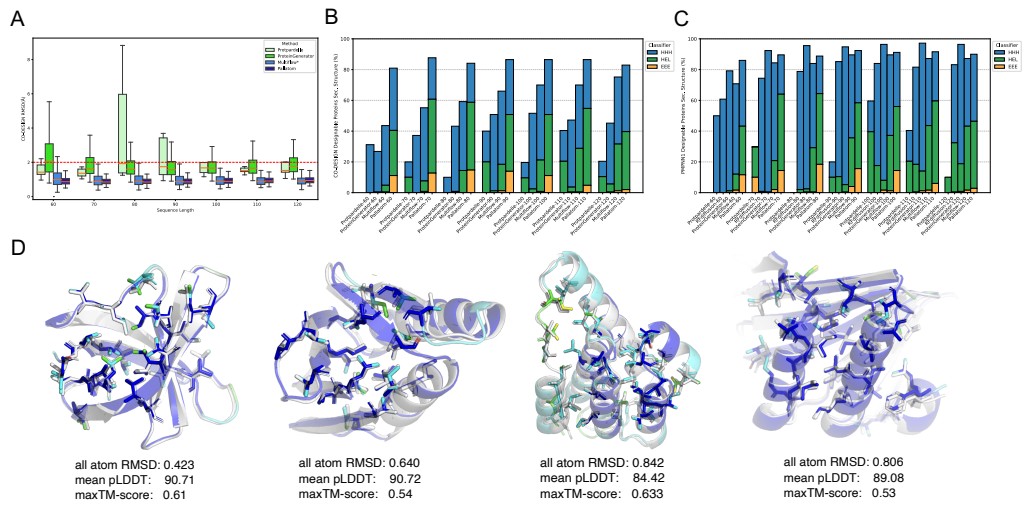

Figure 2: Evaluation of proteins sampled from Pallatom. (A) Boxplot of aaRMSD for proteins sampled by various methods under the CO-DESIGN 1 mode. Multiflow exhibits the $C_\alpha$RMSD. (B, C) The proportions of secondary structures in designable proteins across different lengths are presented for CO-DESIGN 1 and PMPNN 1 modes across various methods, with the total height of the y-axis representing the designability. (D) Examples of high-quality, novel all-atom proteins sampled by Pallatom.

- CO-DESIGN 1: For methods that can predict both all-atom coordinates and sequences, **DES-aa** is used. Diversity and novelty are evaluated based on the all-atom designable proteins.
- PMPNN 1: Other methods use **DES-bb** with $N_{seq} = 1$. Specifically, we calculate and display the two **DES-bb (w/wo)** under conditions with and without the pLDDT > 80 constraint. Diversity and novelty are evaluated based on the proteins filtered by **DES-bb (w)**.

## 4.3 RESULTS

We sample Pallatom with 200 time steps using a noise scale $\gamma_0 = 0.2$, a step scale $\eta = 2.25$ and evaluate 250 proteins sampled for each length $L = 60, 70, 80, 90, 100, 110, 120$. Our primary comparisons are with state-of-the-art methods capable of generating all-atom proteins, such as Protpardelle (Chu et al., 2024) and

ProteinGenerator (Lisanza et al., 2023). For backbone generation, we compare with RFdiffusion (Watson et al., 2023). We also compared Multiflow (Campbell et al., 2024), which is capable of simultaneously generating both backbones and sequences. All methods are evaluated using their open-source code and default parameters.

As shown in Table 1, Pallatom surpasses previous methods in the CO-DESIGN 1 evaluation for generating all-atom proteins. After incorporating pLDDT filter to ensure sequence quality, we observed that sequences generated by Pallatom are comparable to those generated by ProteinMPNN. Remarkably, even though Pallatom has not undergone any training in fixed-backbone design tasks, it achieves this performance by predicting sequences using a single linear layer derived from aggregated atomic-level features. This remarkable achievement stands out as the only high-performance method capable of designing all-atom structure among existing approaches. This supports our hypothesis that learning $P(all\text{-}atom)$ can effectively capture the relationship between protein structure and sequence.

Furthermore, the all-atom protein structures generated by Pallatom show greater structural diversity. With the same number of samples, Pallatom achieves twice the diversity of Multiflow and three times that of ProteinGenerator. This highlights Pallatom's enhanced ability to explore a wider range of conformations. Likewise, Pallatom achieves the highest level of sequence diversity among all existing methods.

In the evaluation of backbone generation, we observed that without the pLDDT quality constraint, the DES-bb metric is always overestimated, as evidenced by the substantial discrepancy between the two designability metrics in Protpardelle for the same task. ProteinGenerator achieves an almost perfect backbone designability, significantly surpassing other models. This indicates that sequence-based generative models can capture highly designable backbone structures. However, its performance in structural diversity falls short compared to structure-based generative models, highlighting a subtle trade-off between designability (DES-bb) and structural diversity (DIV-str). Protpardelle performs below its counterparts across all metrics, suggesting that network architecture and atomic representation play a crucial role in generative capability. Similar to the results of the CO-DESIGN 1 evaluation, Pallatom not only generates protein structures with higher diversity but also maintains strong designability and novelty.

We present a more detailed analysis in Figure 2. Pallatom maintains stable performance across all tested protein lengths, showing a balanced distribution of secondary structures. In the secondary structure distribution of designable proteins under CO-DESIGN 1 and PMPNN 1 illustrated in Figures 2B and 2C, we observe that comparative methods exhibit a limited preference for secondary structures, particularly in failing to generate proteins with predominantly $\beta$-sheet structures. For example, the ProteinGenerator model shows a marked preference for generating proteins with $\alpha$-helical structures. This excessive tendency to produce a single type of secondary structure deviates from our expectations.

Several designable all-atom proteins for case studies in Figure 2D. These highly novel proteins designed by Pallatom show a highly ordered side-chain distribution, with hydrophobic side-chain concentrated internally to form a stable hydrophobic core, and the surface covered by hydrophilic polar residues. This distribution is consistent with the side-chain distribution of high pLDDT structures predicted by ESMFold, demonstrating that Pallatom has learned the physical and chemical properties that govern protein folding and residue distribution from the all-atom distribution.

We further analyzed the performance of Pallatom on out-of-distribution (OOD) lengths. Specifically, we conducted sampling and evaluation for lengths from 150 to 400. The results demonstrate that Pallatom exhibits exceptional scalability, achieving the highest designability even at over twice the maximum training length ($L = 128$). Detailed results are provided in the appendix.

Table 2: Pallatom sample metrics.

| Noise Level $\gamma_0$ | 0.2 | 0.2 | 0.1 | 0.2 | 0.2 |
|---|---|---|---|---|---|
| Step Scale $\eta$ | 1.75 | 2.25 | 2.25 | 2.75 | 3.25 |
| $N_{\text{steps}}$ | 200 | 200 | 200 | 200 | 200 |
| DES-aa ($\uparrow$) | 57% | 87% | 89% | **94%** | 93% |
| DIV-str ($\uparrow$) | 56 | **64** | 52 | 55 | 46 |
| HHH(%) | 28% | 28% | 45% | 35% | 38% |
| HEL(%) | 56% | 54% | 44% | 47% | 51% |
| EEE(%) | 16% | 18% | 11% | 18% | 11% |

## 4.4 HYPERPARAMETER

We analyzed the impact of the sampling parameters and Table 5 presents the metrics for Pallatom when sampling 250 proteins with $L = 100$. We observed that, under the same noise scale, increasing the step scale $\eta$ leads to a corresponding rise in designability, as well as an increase in the proportion of all-helix structures within the secondary structure. However, this improvement in designability comes at the cost of reduced structural diversity, indicating a trade-off between these two metrics. We further demonstrated the impact of varying step scales on the sampling of OOD-length proteins, with detailed results provided in the appendix. Furthermore, we found that reducing the additional noise level to $\gamma = 0.1$ slightly enhances designability but also significantly decreases structural diversity, consistent with the findings of previous work (Yim et al., 2023b).

## 5 DISCUSSION

We introduce Pallatom, a highly efficient end-to-end all-atom protein generation framework that simultaneously captures the relationship between sequence and structure, enabling state-of-the-art performance. Our modification of the `atom14` representation removes the limitations of explicitly defining all amino acid types during generation and provides a more accurate method for representing all-atom system coordinates. We redesign the dual-track architecture of AlphaFold3 diffusion module, by developing a novel mechanism comprising " traversing" representations and dual-track recycling method. The new framework efficiently adapts to all-atom protein structure diffusion generation. The results demonstrate that models learning $P(all\text{-}atom)$ exhibit strong performance and diversity in *de novo* protein generation, unlocking new pathways for protein design. Future work includes developing a more generalized model architecture, expanding our system to support the representation of small molecules, DNA, and non-standard amino acids, and further enhancing the model's capabilities in designing large complex systems, such as antibody complexes and self-assembling materials.

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

## A   RELATED WORK

**Backbone Generation.**   Diffusion models based on SE(3)-equivariant network architectures and protein representations using rigid frames have achieved significant success in protein generation, as evidenced by models like Chroma (Ingraham et al., 2023), Genie2 (Lin et al., 2024), RFdiffusion (Watson et al., 2023), FrameDiff (Yim et al., 2023b), FrameFlow (Yim et al., 2023a), Proteus (Wang et al., 2024), and FoldFlow2 (Huguet et al., 2024).   These models now support multi-condition controlled generation and have been extensively validated through both in-silico and wet-lab experiments.

**Codesign Models.**   Recent methods have explored a co-design approach that simultaneously designs both the backbone and sequence. Multiflow (Campbell et al., 2024) utilizes a diffusion process that jointly operates on discrete sequences and continuous SE(3) backbones, eliminating the need for sequence redesign with ProteinMPNN (Dauparas et al., 2022). CarbonNovo (Ren et al., 2024) employs a similar approach, using SE(3) diffusion on the protein backbone while simultaneously designing a sequence at each step with the MRF decoder. The sequence information is then embedded using the protein language model ESM2-3B (Rives et al., 2021) to guide structural generation.

**All-Atom Generation.**   Recent research teams have begun exploring fully all-atom generative representation systems. For instance, Protpardelle (Chu et al., 2024) employs a coordinates diffusion model to support the generation of all-atom protein structures. ProteinGenerator (Lisanza et al., 2023) applies Euclidean diffusion on one-hot encoded sequences, combined with a structure prediction module to obtain all-atom structure. RFdiffusionAA (Krishna et al., 2024), based on fine-tuning the RoseTTAFold2 (Baek et al., 2023), can produce backbone structures of proteins and small molecule complexes but lacks side-chain conformations for standard amino acids. We focus on the methods directly generate all-atom protein structures, the most relevant work is Protpardelle, which uses an end-to-end approach to generate all-atom structures.

## B   TRAINING DATASETS

### B.1   PDB DATA

We used PISCES (Wang & Dunbrack Jr, 2003) to obtain the required PDB list. For training, we selected a subset of PDB entries with a resolution of $<3$Å and a 95% sequence identity threshold. We then performed standard filtering to remove any proteins with $>50\%$ loops and applied a series of folding quality filters (described below). This process resulted in 7,459 structures.

### B.2   AUGMENTED DATA

Augmented data are widely used in protein modeling-related work. Consequently, we supplemented our dataset with the AlphaFold Database (AFDB) (Varadi et al., 2021). The AlphaFold2 predicted structure database is available under a CC-BY-4.0 license for both academic and commercial uses. The AFDB contains a total of 214 million data points. By applying an average pLDDT threshold of $>80$ and limiting the sequence length to a maximum of 128, we obtained 582,652 structures. In addition, we employed more sophisticated filtering strategies to acquire designable and high-quality data, assessing the average neighboring atomic density for each residue to determine whether they are core residues. We found that with this filter, core residue content $>30\%$ allows us to identify tightly packed structures. Furthermore, the number and distribution of secondary structures can define the compactness and diversity of folding. We used DSSP (Touw et al., 2015) to assign the protein secondary structures and removing structures with loop content $>50\%$. We also apply filter on the total number of continuous segments of $\beta$-sheet and $\alpha$-helix to maintain structural diversity. For highly extended structures, we limited the radius of gyration ($R_g$) to less than 25.0. To avoid overly long continuous unstructured regions within the protein structures, we restricted the maxi-

mum length of each loop to 15. Finally, we used the FoldSeek easycluster algorithm to remove redundant structures, setting a TM score threshold of 0.8 and a coverage of 0.9, which removed approximately 30% of highly similar structures. After applying these stringent filters, only 27,697 protein structures remained. These structures typically exhibit good folding and high designability.

## C ALGORITHMS

We provide a detailed description of the Pallatom modules and algorithm workflow below. In the pseudocode, the algorithms highlighted in blue are nearly identical to those of AlphaFold3 and are not expanded to avoid unnecessary repetition.

### C.1 MAINTRUNK

Algorithm 2 details the denoising process of the Pallatom `MainTrunk` network. Figure 3 displays the detailed computational workflow for the AtomDecoder unit.

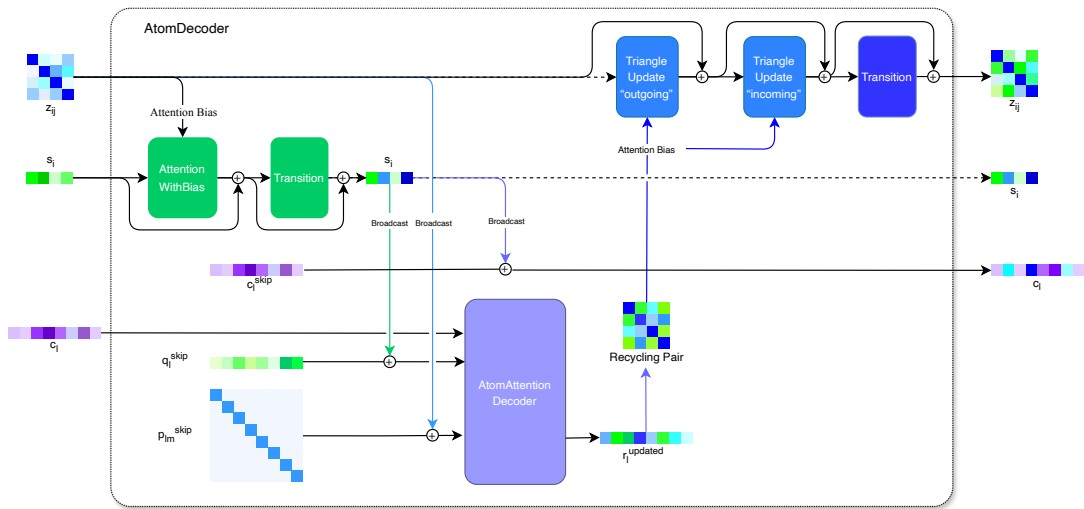

Figure 3: The detail architecture of the AtomDecoder unit.

### C.2 TEMPLATEEMBEDDER

In the TemplateEmbedder module, we retained only the necessary mask and distogram features, and additionally included the timestep feature for self-conditioning.

### C.3 ATOMFEATUREENCODER

The core of the AtomFeatureEncoder is derived from AlphaFold3, and we made two adjustments based on specific tasks. First, To process standard protein residue conformations, we reduced the 20 original amino acid conformations to a single backbone conformation, represented by alanine. Additionally, each residue

---

**Algorithm 2** MainTrunk

---

**def MainTrunk** $(\{f^*\}, \{\mathbf{r}_l^{input}\}, \hat{t}, t, c_{atom} = 128, c_{pair} = 128, c_{token} = 256, c_{atompair} = 16, \sigma_{data} = 16)$:

*# Initialize positions and conditions embedding*

1: $\vec{\mathbf{r}}_l^{\text{scaled}} = \vec{\mathbf{r}}_l^{\text{intput}}/\sqrt{\sigma_{\text{data}}^2 + \hat{t}^2}$ $\qquad\qquad\qquad\qquad\qquad\qquad\qquad\qquad \vec{\mathbf{r}}_l^{\text{scaled}} \in \mathbb{R}^3$

2: $\mathbf{s}_i^{init} = \text{LinearNoBias}((\mathbf{f}_i^{\text{residue\_idx}}))$ $\qquad\qquad\qquad\qquad\qquad\qquad \mathbf{s}_i^{init} \in \mathbb{R}^{c_{token}}$

3: $\mathbf{t}_i = \text{TimeFourierEmbedding}(\frac{1}{4}log(\hat{t}/\sigma_{data}))$ $\qquad\qquad\qquad\qquad \mathbf{t}_i \in \mathbb{R}^{c_{token}}$

4: $\mathbf{s}_i^{init} \mathrel{+}= \mathbf{t}_i$

5: $\mathbf{z}_{ij}^{init} = \text{RelativePositionEncoding}(f^*)$ $\qquad\qquad\qquad\qquad\qquad \mathbf{z}_{ij}^{init} \in \mathbb{R}^{c_{pair}}$

6: $\mathbf{z}_{ij} = \mathbf{z}_{ij}^{init} + \text{TemplateEmbedder}(\{f^*\}, \mathbf{z}_{ij}^{init}, t, N_{block} = 2, c = 64, d = c_{pair})$

*# Initialise single and atom embeddings*

7: $\{\mathbf{s}_i\}, \{\vec{\mathbf{q}}_l^{skip}\}, \{\mathbf{c}_l^{skip}\}, \{\mathbf{p}_{lm}^{skip}\}, \{\mathbf{c}_l\} = \text{AtomFeatureEncoder}(\{f^*\}, \mathbf{s}_i^{init}, \mathbf{z}_{ij}, \vec{\mathbf{r}}_l^{\text{scaled}}, c_{token}, c_{atompair}, c_{atom})$

8: $\mathbf{s}_i \mathrel{+}= \text{LinearNoBias}(\text{LayerNorm}(\mathbf{s}_i^{init}))$

*# AtomDecoder Units*

9: $\vec{\mathbf{r}}_l^{\text{updates}} = \mathbf{0}$ $\qquad\qquad\qquad\qquad\qquad\qquad\qquad\qquad\qquad\qquad \vec{\mathbf{r}}_l^{\text{updates}} \in \mathbb{R}^3$

10: **for all** $unit \in \{1, \ldots, K_{\text{unit}}\}$ **do**

11: $\qquad \mathbf{s}_i = \text{NodeUpdate}(\mathbf{s}_i, \mathbf{t}_i, \mathbf{z}_{ij}, c = c_{token})$

12: $\qquad \{\mathbf{q}_l^{updated}\}, \{\vec{\mathbf{r}}_l^{\text{update}}\}, \{\mathbf{c}_l\} = \text{AtomAttentionDecoder}(\mathbf{q}_l^{skip}, \mathbf{p}_{lm}^{skip}, \mathbf{c}_l^{skip}, \mathbf{c}_l, \mathbf{s}_i, \mathbf{z}_{ij})$

13: $\qquad \vec{\mathbf{r}}_l^{\text{updates}} \mathrel{+}= \vec{\mathbf{r}}_l^{\text{update}}$

14: $\qquad \vec{\mathbf{r}}_l^{denoised} = \sigma_{\text{data}}^2/(\sigma_{\text{data}}^2 + \hat{t}^2) \cdot \vec{\mathbf{r}}_l^{\text{intput}} + \sigma_{\text{data}} \cdot \hat{t}/\sqrt{\sigma_{\text{data}}^2 + \hat{t}^2} \cdot \vec{\mathbf{r}}_l^{\text{updates}}$

15: $\qquad \vec{\mathbf{r}}_i^{center} = \vec{\mathbf{r}}_l^{denoised}[center\_uid]$

16: $\qquad \mathbf{z}_{ij} = \text{PairUpdate}(\mathbf{z}_{ij}, \vec{\mathbf{r}}_i^{center}, c = c_{pair})$

17: **end for**

*# SeqHead for amino acid decoding*

18: $\mathbf{a}_i = \underset{\substack{l \in \{1, \ldots, N_{\text{atoms}}\} \\ \text{tok\_idx}(l)=i}}{\text{mean}} \left( \text{ReLU}(\text{LinearNoBias}(\{\mathbf{q}_l^{updated}\})) \right)$ $\qquad\qquad \mathbf{a}_i \in \mathbb{R}^{c_{token}}$

19: $\mathbf{f}_i^{\text{seq\_logits}} = \text{LinearNoBias}(\mathbf{a}_i)$ $\qquad\qquad\qquad\qquad\qquad\qquad \mathbf{f}_i^{\text{seq\_logits}} \in \mathbb{R}^{20}$

**return** $\{\vec{\mathbf{r}}_l^{denoised}\}, \{\mathbf{f}_i^{\text{seq\_logits}}\}$

---

**Algorithm 3** Template Embedder

---

**def TemplateEmbedder** $(\{f^*\}, \{\mathbf{z}_{ij}\}, t, N_{block} = 2, c = 64, d = 128)$:

*# Concat template features*

1: $b_{ij}^{\text{template\_pseudo\_beta\_mask}} = f_i^{\text{template\_pseudo\_beta\_mask}} \cdot f_j^{\text{template\_pseudo\_beta\_mask}}$

2: $b_{ij}^{\text{time}} = t \odot f_{ij}^{\text{template\_pseudo\_beta\_mask}}$ $\qquad\qquad\qquad\qquad\qquad\qquad t \sim [0, 1)$

3: $\mathbf{a}_{ij} = \text{concat}(f_{ij}^{\text{template\_distogram}}, b_{ij}^{\text{template\_pseudo\_beta\_mask}}, b_{ij}^{\text{time}})$

*# Embed self-condition positions*

4: $\mathbf{v}_{ij} = \text{LinearNoBias}(\text{LayerNorm}(\mathbf{z}_{ij})) + \text{LinearNoBias}(\mathbf{a}_{ij})$ $\qquad\qquad \mathbf{v}_{ij} \in \mathbb{R}^c$

5: $\mathbf{v}_{ij} = \text{PairformerStack}(\mathbf{v}_{ij}, N_{block})$

6: $\mathbf{u}_{ij} \leftarrow \text{LinearNoBias}(\text{ReLU}(\text{LayerNorm}(\mathbf{v}_{ij})))$ $\qquad\qquad\qquad \mathbf{u}_{ij} \in \mathbb{R}^d$

**return** $\{\mathbf{u}_{ij}\}$

---

(token) was standardized to have 14 atoms, represented in the `atom14` format. To prevent information leakage from the virtual atoms, they were defined based on the $C_\alpha$ atom. Secondly, to accommodate the recycling update mechanism between blocks, we utilized traversing atomic-level representations, these features flow between blocks and only integrate with token-level information during the decoding phase.

---

**Algorithm 4** AtomFeatureEncoder

---

**def AtomFeatureEncoder** $(\{f^*\}, \mathbf{s}_i^{input}, \mathbf{z}_{ij}^{input}, \vec{\mathbf{r}}_l^{scaled}, c = 256, d = 16, m = 128)$:

*# Create the atom single conditioning: weighted by sequence*

1: $\mathbf{f}^{ref} = \text{concat}(\vec{\mathbf{f}}^{\text{ref\_pos}}, \mathbf{f}^{\text{ref\_element}})$           $\mathbf{f}^{ref} \in \mathbb{R}^{14 \times 7}$

2: $\mathbf{c}_l = \text{LinearNoBias}(\text{tile}(\mathbf{f}^{ref}))$           $\mathbf{c}_l \in \mathbb{R}^m$

3: $\vec{\mathbf{f}}_l^{\text{ref\_pos}} = \text{tile}(\vec{\mathbf{f}}^{\text{ref\_pos}})$           $\vec{\mathbf{f}}_l^{\text{ref\_pos}} \in \mathbb{R}^3$

4: $\mathbf{c}_l^{skip} = \mathbf{c}_l$

*# Embed offsets between atom reference positions*

5: $\vec{\mathbf{d}}_{lm} = \vec{\mathbf{f}}_l^{\text{ref\_pos}} - \vec{\mathbf{f}}_m^{\text{ref\_pos}}$           $\vec{\mathbf{d}}_{lm} \in \mathbb{R}^3$

6: $v_{lm} = (\mathbf{f}_l^{\text{ref\_space\_uid}} == \mathbf{f}_m^{\text{ref\_space\_uid}})$           $v_{lm} \in \mathbb{R}$

7: $\mathbf{p}_{lm} = \text{LinearNoBias}(\vec{\mathbf{d}}_{lm}) \cdot v_{lm}$           $\mathbf{p}_{lm} \in \mathbb{R}^d$

8: $\mathbf{p}_{lm} \mathrel{+}= \text{LinearNoBias}\left(1/(1 + \|\vec{\mathbf{d}}_{lm}\|^2)\right) \cdot v_{lm}$

9: $\mathbf{p}_{lm} \mathrel{+}= \text{LinearNoBias}(v_{lm}) \cdot v_{lm}$

10: $\mathbf{p}_{lm} \mathrel{+}= \text{LinearNoBias}(\text{ReLU}(\mathbf{c}_l)) + \text{LinearNoBias}(\text{ReLU}(\mathbf{c}_m))$

11: $\mathbf{p}_{lm}^{skip} = \mathbf{p}_{lm}$

*# Initialise the atom single representation as the single conditioning*

12: $\mathbf{q}_l^{skip} = \mathbf{c}_l + \text{LinearNoBias}(\vec{\mathbf{r}}_l^{scaled})$           $\mathbf{q}_l \in \mathbb{R}^m$

*# Add atom positional and time conditioning*

13: $\mathbf{c}_l \mathrel{+}= \text{LinearNoBias}(\text{LayerNorm}(\mathbf{s}_{\text{tok\_idx}(l)}^{init}))$

14: $\mathbf{p}_{lm} \mathrel{+}= \text{LinearNoBias}(\text{LayerNorm}(\mathbf{z}_{\text{tok\_idx}(l)\text{tok\_idx}(m)}^{init}))$

15: $\mathbf{p}_{lm} \mathrel{+}= \text{LinearNoBias}(\text{ReLU}(\text{LinearNoBias}(\text{ReLU}(\text{LinearNoBias}(\text{ReLU}(\mathbf{p}_{lm}))))))$

*# Cross attention transformer*

16: $\mathbf{q}_l^{skip} = \text{AtomTransformer}(\mathbf{q}_l^{skip}, \mathbf{c}_l, \mathbf{p}_{lm}, N_{\text{block}} = 3, N_{\text{head}} = 4)$

*# Aggregate per-atom representation to per-token representation*

17: $\mathbf{s}_i = \underset{\substack{l \in \{1, \ldots, N_{\text{atoms}}\} \\ \text{tok\_idx}(l) = i}}{\text{mean}} \left(\text{ReLU}(\text{LinearNoBias}(\mathbf{q}_l^{skip}))\right)$           $\mathbf{s}_i \in \mathbb{R}^c$

**return** $\{\mathbf{s}_i\}, \{\mathbf{q}_l^{skip}\}, \{\mathbf{c}_l^{skip}\}, \{\mathbf{p}_{lm}^{skip}\}, \{\mathbf{c}_l\}$

---

### C.4 ATOMATTENTIONDECODER

We established a method for managing token-level and atomic-level information in the decoder layer. The updated single- and pair-representations are simultaneously injected into the traversing atomic-level representations, effectively preventing the repeated accumulation of redundant structural information.

### C.5 NODEUPDATE

In AlphaFold3's diffusion module, the single representation is updated by both the AttentionPairBias and ConditionTransition algorithms, which use pre-defined structural representations as conditions. The AttentionPairBias algorithm, which can be viewed as a conditional version of the RowAttentionWithBias algorithm from AlphaFold2, additionally employs Adaptive LayerNorm (Ba, 2016) and SwiGLU (Shazeer, 2020)

---

**Algorithm 5** AtomAttentionDecoder

---

**def AtomAttentionDecoder** $(\mathbf{q}_l^{skip}, \mathbf{p}_{lm}^{skip}, \mathbf{c}_l^{skip}, \mathbf{c}_l, \mathbf{s}_i, \mathbf{z}_{ij})$:

*# Add trunk embeddings*

1: $\mathbf{q}_l = \text{LinearNoBias}(\text{LayerNorm}(\mathbf{s}_{\text{tok\_idx}(l)})) + \mathbf{q}_l^{\text{skip}}$

2: $\mathbf{p}_{lm} = \text{LinearNoBias}(\text{LayerNorm}(\mathbf{z}_{\text{tok\_idx}(l)\text{tok\_idx}(m)}) + \mathbf{p}_{lm}^{\text{skip}}$

3: $\mathbf{p}_{lm} \mathrel{+}= \text{LinearNoBias}(\text{ReLU}(\text{LinearNoBias}(\text{ReLU}(\text{LinearNoBias}(\text{ReLU}(\mathbf{p}_{lm}))))))$

*# Cross attention transformer*

4: $\mathbf{q}_l = \text{AtomTransformer}(\mathbf{q}_l, \mathbf{c}_l, \mathbf{p}_{lm}, N_{\text{block}} = 3, N_{\text{head}} = 4)$

*# Map to positions update*

5: $\vec{\mathbf{r}}_l^{\text{update}} = \text{LinearNoBias}(\text{LayerNorm}(\mathbf{q}_l))$

*# Update trunk embeddings to atom condition*

6: $\mathbf{c}_l = \text{LinearNoBias}(\text{LayerNorm}(\mathbf{s}_{\text{tok\_idx}(l)})) + \mathbf{c}_l^{\text{skip}}$

**return** $\{\vec{\mathbf{r}}_l^{\text{update}}\}, \{\mathbf{c}_l\}$

---

techniques to conditionally scale the representation values. The ConditionTransition algorithm acts as a gate. Due to the lack of structural condition representation in generative models, we replace the structural condition with time-based condition in the AttentionPairBias. This allows for adaptive scaling of network updates based on the level of noise. Furthermore, we considered updating the single information between blocks using residual connections instead of ConditionTransition. Our preliminary tests indicate that this update method prevents the degradation of the single representation. We also used dropout during training to mitigate the risk of overfitting and enhance the model's robustness.

---

**Algorithm 6** NodeUpdate

---

**def NodeUpdate** $(\mathbf{s}_i, \mathbf{t}_i, \mathbf{z}_{ij}, c = 256)$:

*# AttentionPairBias with updated pair bias*

1: $\mathbf{s}_i \mathrel{+}= \text{DropoutRowwise}_{0.25}(\text{AttentionPairBias}(\mathbf{s}_i, \mathbf{t}_i, \mathbf{z}_{ij}, \beta_{ij} = 0, N_{head} = 8))$

2: $\mathbf{s}_i \mathrel{+}= \text{Transition}(\mathbf{s}_i)$

**return** $\{\mathbf{s}_i\}$

---

### C.6 PAIRUPDATE

In the PairUpdate module, to minimize the number of parameters and maximize network depth, we opted to use only the TriangleAttention algorithm, omitting TriangleMultiplication. Ablation studies from AlphaFold2 suggest that relying solely on TriangleAttention still enables accurate protein structure prediction. We used a modified TriangleAttention algorithm, which uses the pair representation from the recycling structure as attention bias. To maintain consistency in the pair feature space, we binned the recycling structure using the same parameters as in the TemplateEmbedder module, with a total of 39 bins ranging from 3.25 to 50.75 Å.

### C.7 SMOOTH LDDT LOSS FUNCTION

During the training, we adopt the smooth LDDT loss as proposed by AlphaFold3, and implement a simplified version specifically for all-atom proteins.

---

**Algorithm 7** PairUpdate

---

**def PairUpdate** $(\mathbf{z}_{ij}, \mathbf{r}_i^{center}, c = 128)$:
*# Obtaining the pairwise distance matrix through RBF discretization*
1: $\mathbf{d}_{ij} = \left\| \vec{\mathbf{r}}_i^{center} - \vec{\mathbf{r}}_j^{center} \right\|$            $\mathbf{d}_{ij} \in \mathbb{R}$
2: $\mathbf{b}_{ij} = \text{LinearNoBias}(\text{Transform\_RBF}(\mathbf{d}_{ij}))$           $\mathbf{b}_{ij} \in \mathbb{R}^c$
*# TriangleAttention with coordinates pair bias*
3: $\mathbf{z}_{ij} \mathrel{+}= \text{DropoutRowwise}_{0.25}(\text{TriangleAttentionStartingNodeWithBias}(\mathbf{z}_{ij}, \mathbf{b}_{ij}))$
4: $\mathbf{z}_{ij} \mathrel{+}= \text{DropoutColumnwise}_{0.25}(\text{TriangleAttentionEndingNodeWithBias}(\mathbf{z}_{ij}, \mathbf{b}_{ij}))$
5: $\mathbf{z}_{ij} \mathrel{+}= \text{Transition}(\mathbf{z}_{ij})$
**return** $\{\mathbf{z}_{ij}\}$

---

**Algorithm 8** Smooth LDDT loss

---

**def SmoothLDDTLoss** $(\vec{\mathbf{r}}_l, \vec{\mathbf{r}}_l^{GT})$:
*# Compute distances between all pairs of atoms*
1: $\delta r_{lm} \leftarrow \| \vec{\mathbf{r}}_l - \vec{\mathbf{r}}_m \|$
2: $\delta r_{lm}^{GT} \leftarrow \| \vec{\mathbf{r}}_l^{GT} - \vec{\mathbf{r}}_m^{GT} \|$
*# Compute distance difference*
3: $\delta_{lm} = \text{abs}(\delta r_{lm}^{GT} - \delta r_{lm})$
4: $\epsilon_{lm} = \frac{1}{4}[\text{sigmoid}(\frac{1}{2} - \delta_{lm}) + \text{sigmoid}(1 - \delta_{lm}) + \text{sigmoid}(2 - \delta_{lm}) + \text{sigmoid}(4 - \delta_{lm})]$
*# Set the radius threshold and compute mean*
5: $c_{lm} \leftarrow \mathbf{1}(\delta x_{lm}^{GT} < 15\text{Å})$
5: $\text{lddt} = \underset{l \neq m}{\text{mean}}(c_{lm}\epsilon_{lm}) / \underset{l \neq m}{\text{mean}}(c_{lm})$
**return** 1 - lddt

---

Table 3: Input Feature Descriptions

| Input Feature | Dimension | Description |
|---|---|---|
| ref_pos | $(14, 3)$ | Atom positions in the reference conformer are given in Å. The backbone atoms (N, C, $C_\alpha$, O, $C_\beta$) are listed in the first five columns, while all side-chain atoms are moved to the $C_\alpha$ atom position. |
| ref_element | $(14, 4)$ | We encode the backbone atoms based on their elemental types [N, C, O], while the side-chain atoms are encoded as a single class using 'UNK' (unknown). |
| ref_space_uid | $(N_{atom},)$ | Numerical encoding of the residue index associated with this reference conformer. |
| ref_center_mask | $(N_{atom},)$ | Masks indicating the center atom of the residue. |
| residue_index | $(N_{token},)$ | The pdb residue number for calculating relative positional embedding |
| residx_embedding | $(N_{token}, 32)$ | The absolute position embedding by sinusoidal positional encoding. |
| template_distogram | $(N_{token}, N_{token}, 39)$ | Pairwise distogram of pseudo $C_\beta$ are discretized into 38 bins of equal width between of bin_min=3.25Å, bin_max=50.75Å, one more bin contains any larger distances. |
| template_cb_mask | $(N_{token},)$ | Mask indicating if the $C_\beta$ atom has coordinates for the template at this residue, where 1 indicates existing tokens and 0 is used for padding tokens. |
| template_time | $(N_{token}, N_{token}, 1)$ | Normalized pairwise time-step feature, ranging from 0 to 1. |
| all_atom_positions | $(N_{atom}, 3)$ | The noisy position of all atoms in the system. |
| all_atom_mask | $(N_{atom},)$ | Mask indicating which atom slots are used in the the system. |
| t_hat | $(1,)$ | The noise level value for adding noise. |

## D  EXPERIMENT DETAILS

### D.1  MODEL DETAILS

Table 4 provides a detailed list of the hyperparameters used for training.

Table 4: Pallatom training hyperparameters.

| Parameter name | Value |
|---|---|
| Batch size | 32 |
| Learning rate | 0.001, No warm-up. |
| Examples per epoch | 35156 |
| Crop size | 128 |
| Loss weights | Sequence loss weight $\alpha_0 = 0.25$,
Smooth lddt loss weight $\alpha_1 = 1.0$,
Token-level distogram loss weight $\alpha_2 = 0.5$,
Atomic-level distogram loss weight $\alpha_3 = 0.5$,
Intermediate loss weight $\alpha_4 = 1.0$
In the basic loss $\mathcal{L}_0$, weight allocation was applied to residue types, with a weight of 2.0 assigned to polar residues, and 1.0 to the others. |
| Diffusion timesteps ($N_{steps}$ or $T$) | 200 |
| Self-condition rate | 100% |
| EDM | Noise schedule `lognormal`. $\ln(\sigma) \sim \mathcal{N}(P_{mean}, P_{std}^2)$, $P_{mean} = -1.2$, $P_{std} = 1.5$, $\sigma_{data} = 16$,
Stochastic sampler $t_{min} = 0.01$, $t_{max} = 1.0$, noise level $\gamma_0 = 0.2$, noise scale $\lambda = 1.003$, step scale $\eta = 2.25$ |
| Transformer | single representation dimension = 256, pair representation dimension=128, number of heads = 8, number of decoder units = 8 |
| Training Steps | $3 \times 10^5$ |
| Training time | $\approx 10$ days |
| Device | $4 \times$ A6000 |

### D.2  ADDITIONAL RESULTS

#### D.2.1  OUT-OF-DISTRIBUTION PERFORMANCE

**Evaluation of Metrics**  We conducted a comparative evaluation of Pallatom on longer sequence lengths not encountered during training. Specifically, we sampled 250 proteins for each of the lengths $L = 150, 200, 250, 300, 350, 400$ for assessment. Notably, the maximum sequence length in Pallatom's training set was 128, whereas in all comparison methods, the training data includes proteins with a maximum length of up to 384. Figures 4 illustrate the designability, structural diversity, and structural novelty for each length under the CO-DESIGN 1 and PMPNN 1 modes, respectively.

We observed that in both evaluation modes, Pallatom exhibited the highest designability below twice the maximum training length ($L = 150 - 250$), with structural diversity and novelty significantly surpassing the Multiflow. At $L = 300$, Pallatom demonstrated designability comparable to Multiflow while outperforming it in diversity and novelty. Even when extending to more than three times the maximum training length at $L = 350$ and $L = 400$, Pallatom, although less advantageous in backbone design, still maintained the ability to generate all-atom proteins, a feat unachievable by the other two comparison methods, Protpardelle and ProteinGenerator.

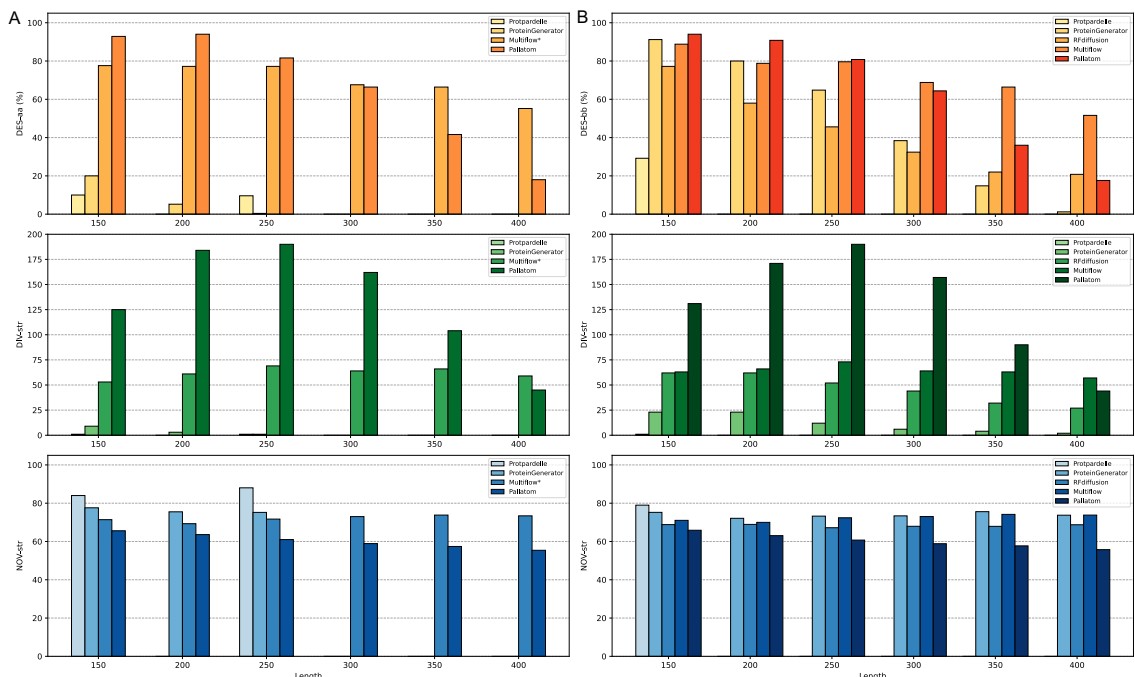

Figure 4: Comparison of Evaluation Metrics for Sampled Proteins at Longer Lengths. (A) and (B) respectively show the designability, structural diversity, and structural novelty under the CO-DESIGN 1 mode and the PMPNN 1 mode.

**Secondary Structure Analysis** We further analyzed the secondary structure preferences of sampled structures from all methods. Specifically, we utilized DSSP to classify the secondary structure of each residue in the proteins. If the number of $\alpha$-helix residues is more than five times the number of $\beta$-sheet residues, the protein is classified as **HHH**, indicating an all-helix structure. Conversely, if the number of $\beta$-sheet residues exceeds five times the number of helix residues, the protein is classified as **EEE**, indicating an all-$\beta$-sheet structure. In other cases, where the proportions of the two secondary structures are balanced, the protein is classified as **HEL**, representing a $\alpha\beta$ mixed structure.

Figure 5 shows the secondary structure distributions of each method in the two evaluation modes, as well as the secondary structure distributions of the designable proteins. This result indicates the ProteinGenerator, multiflow, and Pallatom exhibit similar secondary structure preferences within the length range of 150-400, with a roughly equal distribution between **HEL** and **HHH** structures. In contrast, RFdiffusion and

Protpardelle show a stronger preference for HEL structures. Within the 150-400 length range, all models rarely succeed in generating **EEE** structures.

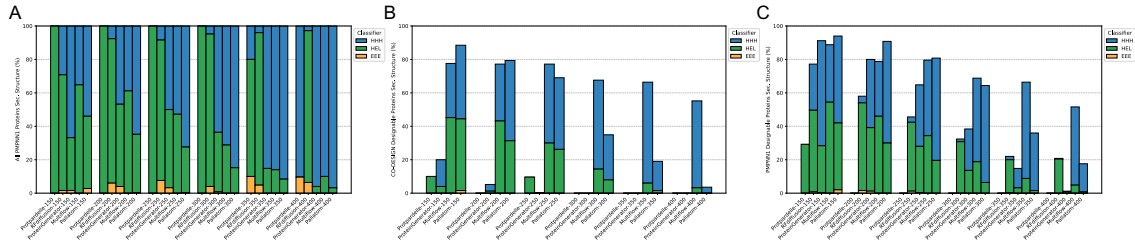

Figure 5: Secondary Structure Distribution of Sampled Proteins at Longer Lengths. Figures (A), (B), and (C) show the secondary structure distribution of all sampled proteins across all methods, the designable proteins in CO-DESIGN 1 mode, and the designable proteins in PMPNN 1 mode, respectively.

All these experimental results demonstrate the superiority of the Pallatom model framework, highlighting its remarkable scalability and generalization capabilities. We present additional case studies of proteins sampled by Pallatom. Figure 7 presents additional examples of novel designable proteins sampled by Pallatom. Figure 8 illustrates the high-quality designable proteins sampled by Pallatom under length distributions not included in the training set.

**Analysis of Sampling Hyperparameters**   We analyzed the effect of the step scale $\eta$ on sampling proteins of unseen longer lengths. In Table 5, the left column for each length corresponds to $\eta = 2.5$, while the right column corresponds to $\eta = 3.0$. We observed that a larger step size in the gradient update direction improves the designability of proteins, and as the number of designable samples increases, the structural diversity of generated proteins also increases, with only a slight decrease in novelty. Additionally, consistent with the observations in the main text regarding the impact of sampling hyperparameters on secondary structure distribution, a larger step size is associated with a more pronounced preference for all-helix structures.

Table 5: Pallatom sample metrics with step scale $\eta = 2.5$ (left) and $\eta = 3.0$ (right).

| Length | 150 | | 200 | | 250 | | 300 | | 350 | | 400 | |
|---|---|---|---|---|---|---|---|---|---|---|---|---|
| DES-aa | 88% | **93%** | 79% | **93%** | 69% | **81%** | 35% | **66%** | 19% | **41%** | 4% | **18%** |
| DIV-str | **131** | 125 | 153 | **184** | 149 | **190** | 73 | **162** | 38 | **104** | 9 | **45** |
| NOV-str | **0.650** | 0.656 | **0.618** | 0.636 | **0.594** | 0.610 | **0.576** | 0.589 | **0.551** | 0.574 | **0.525** | 0.554 |
| HHH(%) | 44% | 53% | 53% | 64% | 54% | 70% | 60% | 77% | 64% | 85% | 69% | 92% |
| HEL(%) | 53% | 44% | 47% | 36% | 46% | 30% | 40% | 23% | 36% | 15% | 31% | 8% |
| EEE(%) | 2.8% | 2.7% | 0.0% | 0.4% | 0.0% | 0.4% | 0.0% | 0.0% | 0.0% | 0.0% | 0.0% | 0.0% |

### D.2.2   SEQUENCE QUALITY OF DESIGNABLE PROTEINS

We compared the quality of the sequences generated by Pallatom with those produced by ProteinMPNN for the same protein structures designed by Pallatom. Figure 6 shows the pLDDT scores of the two sequences predicted by ESMfold. We found that the sequence confidence score of Pallatom is slightly lower than that of ProteinMPNN, with the maximum mean pLDDT difference not exceeding 2. We attribute this difference to both the training data and the tasks. Regarding training data, Pallatom was trained on a monomer protein dataset, whereas ProteinMPNN was trained on a more diverse dataset that includes both monomer and multichain structures. Additionally, when preparing the training set, ProteinMPNN focused more on the

sequence diversity under the same structure, while Pallatom needed to consider both sequence and structure diversity. In terms of training tasks, the objectives of the two models are fundamentally different. Protein-MPNN is concerned solely with sequence design given a real backbone, whereas Pallatom must balance the dual objectives of structure generation and sequence generation from pure noise.

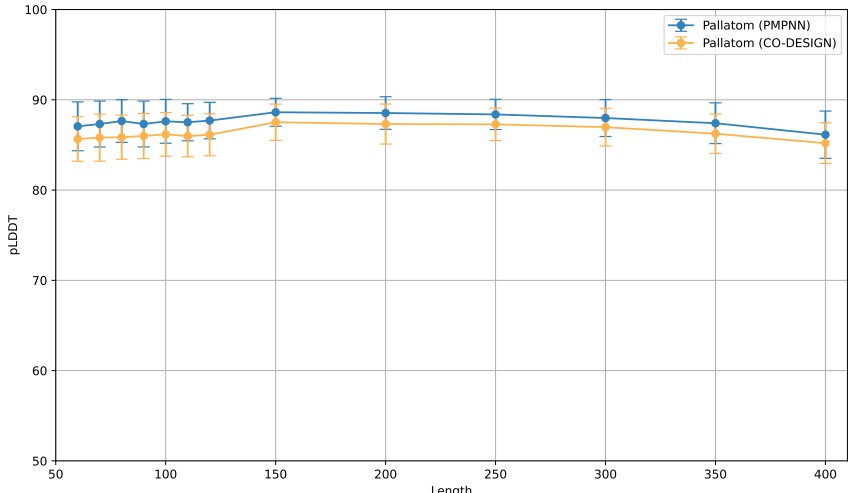

Figure 6: Comparison of pLDDT between sequences designed by Pallatom and ProteinMPNN across different lengths. Sequences designed by Pallatom are labeled as "Pallatom (CO-DESIGN)," while sequences designed by ProteinMPNN based on the backbone are labeled as "Pallatom (PMPNN)."

### D.2.3 ANALYSIS OF SAMPLING TIME

We conducted a comparative analysis of sampling times for each method. Specifically, we standardize the diffusion sampling steps to $T = 200$ and sample 100 proteins for each length, calculating the mean and standard deviation. All methods were tested on the same hardware: CPU: AMD EPYC 7402 @2.8GHz, GPU: NVIDIA GeForce RTX 4090 with 24GB VRAM.

Table 6 presents the results. Thanks to JAX's JIT compilation and our optimizations at the atom level of Attention, Pallatom achieved the second fastest sampling speeds for lengths ranging from 100 to 350, outperforming all methods except Protpardelle. At $L = 400$, even with the atomic-level length reaching $(14 \times 400) \times (14 \times 400)$, Pallatom's performance remains comparable to the second-fastest method, Multiflow, and is 5 times faster than RFdiffusion and 16 times faster than ProteinGenerator.

Table 6: Sampling Time (in seconds). The shortest time is highlighted in bold, and the second shortest is indicated in italics.

| Length | 100 | 150 | 200 | 250 | 300 | 350 | 400 |
|---|---|---|---|---|---|---|---|
| Protpardelle | *10.9±0.1* | **11.3±0.2** | **11.7±0.2** | **12.5±0.2** | **24.5±0.5** | **26.3±0.6** | **27.6±0.9** |
| ProteinGenerator | 414.1±107.5 | 389.5±2.3 | 388.8±1.3 | 477.6±1.7 | 624.0±4.1 | 796.3±4.6 | 950.3±5.3 |
| Multiflow | 25.3±0.4 | 25.3±0.6 | 27.1±0.3 | 29.4±0.2 | 35.1±0.6 | 40.5±0.2 | *46.6±0.5* |
| RFdiffusion | 95.5±14.6 | 93.9±1.0 | 106.3±0.9 | 138.5±0.8 | 183.0±0.8 | 230.7±1.6 | 287.4±6.5 |
| Pallatom* | **10.2±0.1** | *12.1±0.1* | *18.2±0.1* | *21.9±0.1* | *33.3±0.1* | *40.4±0.1* | 57.5±0.1 |

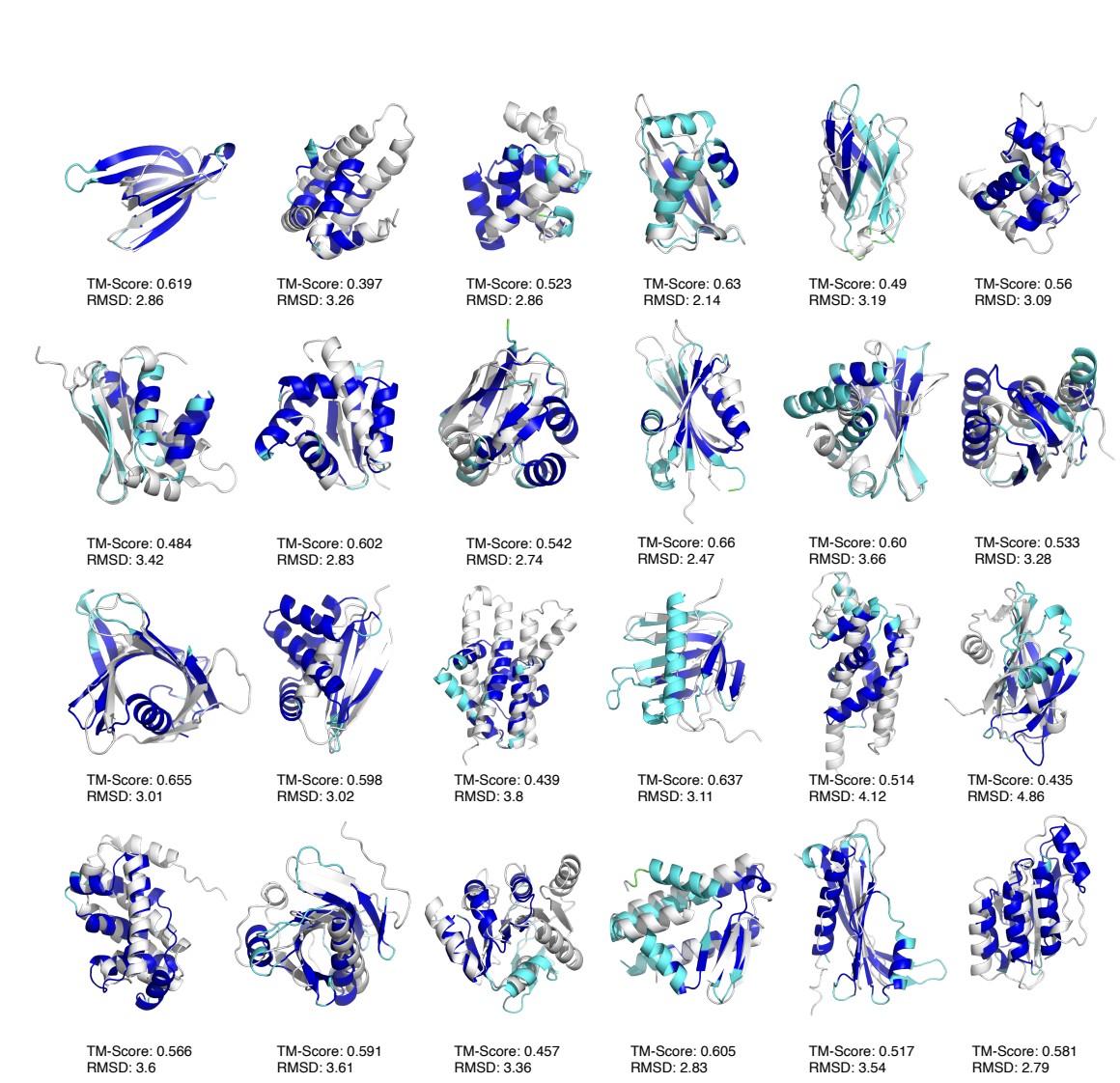

Figure 7: Additional novel designable proteins generated by Pallatom. The blue structures in the figure represent the designable protein sequences generated by Pallatom, which have been predicted using ESMfold and colored based on pLDDT scores. The white structures are the nearest neighbors from the Foldseek database (using the default eight databases on the Foldseek web server), with the distances between the two sets of structures evaluated using TM-Score and RMSD.

L=150

L=200

L=250

L=300

L=350

L=400

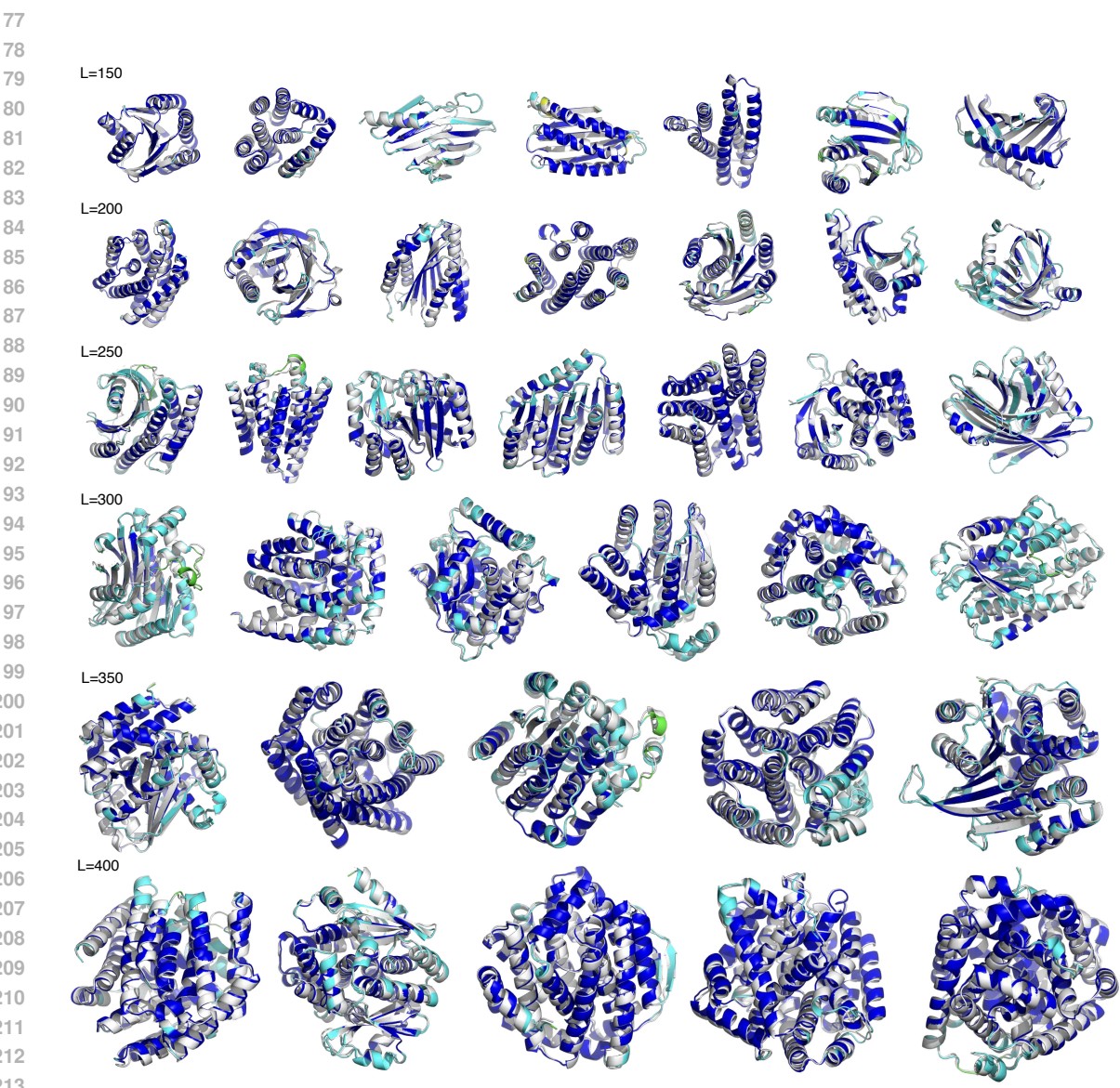

Figure 8: Additional all-atom protein structures generated by Pallatom for longer sequence lengths. The white structures represent those generated by Pallatom, while the colored structures are predicted by ESM-fold and are colored according to pLDDT values, with bluer hues indicating higher pLDDT scores.

