# OpenReview forum: "P(all-atom) Is Unlocking New Path For Protein Design"
_ICLR.cc/2025/Conference — ICLR 2025 Conference Withdrawn Submission_

### Official Review · Reviewer_oX8v · 2024-10-28

**Soundness:** 3
**Presentation:** 3
**Contribution:** 2
**Rating:** 6
**Confidence:** 4

**Summary:**

The paper introduces Pallatom, a generative model for creating all-atom protein structures, aiming to directly model the joint distribution of protein structure and sequence. Pallatom utilizes a novel dual-track architecture that combines atomic- and token-level representations, which are refined iteratively through a multi-layer decoding process. The model introduces the atom14 representation, allowing accurate depiction of side-chain configurations while addressing unknown side-chain coordinates. Pallatom demonstrates high performance in metrics such as designability, diversity, and novelty, positioning it as a promising tool for de novo protein design and efficient generation of complex protein structures.

**Strengths:**

Quality
The paper is methodologically rigorous, presenting a well-designed and thoroughly tested model. Pallatom’s dual-track framework is detailed with an iterative decoding and recycling mechanism that leverages both local and global attention, maintaining consistency between atomic- and token-level representations. The model demonstrates strong results across several key metrics for protein generation—designability, diversity, and novelty—indicating a high-quality, comprehensive evaluation. The detailed ablation studies, use of both PDB and AlphaFold databases for training, and comparisons with several state-of-the-art models highlight the thoroughness of the experimental design. The adoption of diffusion models for sampling noise levels further enhances the robustness and adaptability of the model.

Clarity
The paper is largely clear, particularly in how it describes the architecture and methods of Pallatom. The introduction effectively frames the motivation behind the study, distinguishing Pallatom from previous stepwise or co-design approaches, and introduces the challenges of protein generation concisely. Figures, such as those depicting Pallatom’s architecture, provide useful visual aids, enhancing the reader's understanding of the dual-track representation and multi-layer decoding. However, some sections, such as the mathematical descriptions of the diffusion model and recycling mechanism, could benefit from additional explanation for readers less familiar with advanced protein modeling or generative techniques.

Significance
Pallatom's potential impact on de novo protein design and drug discovery is substantial, especially given the model’s ability to produce highly accurate all-atom structures without requiring iterative backbone and sequence optimization steps. Its emphasis on producing designable and diverse structures makes it a practical tool for real-world applications, where structural accuracy and diversity are key to designing proteins for specific functions. Moreover, the model’s scalability to larger protein lengths without sacrificing accuracy is particularly significant for the study of large, complex protein systems and could pave the way for future expansion to include small molecules or non-standard amino acids.

**Weaknesses:**

1. Limited Interpretability of the Model
While Pallatom achieves strong performance, the model’s interpretability remains limited, which can be a barrier for users seeking insights into how predictions are generated. This is particularly relevant in protein design, where understanding which features drive structure generation is essential for real-world applications.
2. Scalability Constraints on Complex Systems
Pallatom’s training and testing focus largely on shorter protein sequences (up to length 128), and although it exhibits scalability for slightly longer sequences, its practical applicability to more complex protein systems, such as large complexes or multi-chain proteins, remains unclear.
3. Atom14 Representation Constraints
Although the atom14 representation improves handling of unknown side-chain conformations, it is a simplified model that may not capture all physical details, especially for less common or modified amino acids. This could limit Pallatom’s effectiveness in fine-grained applications like enzyme engineering or ligand binding, where side-chain interactions are critical.

**Questions:**

1. The authors state that the atom14 representation is used to avoid potential conflicts arising from the simultaneous design of sequence and structure, but they do not elaborate on the specific nature of these conflicts.  The authors should explain why padding the initial protein with L residues as x = {xl}Ll=1 → x0, considering it as a 3D point cloud distribution P (x0) ∈ RL×14×3, is necessary to address these conflicts.
2. The authors mention that the dual-track framework is inspired by AlphaFold3, but they do not fully explain the rationale behind their specific modifications to the framework for protein generation. The authors should explain how the traversal atomic-level representations help to prevent the repeated accumulation of redundant structural information.  The authors could also provide a more detailed explanation of how the dual-track recycling mechanism works and why it is effective.  For example, the authors could provide more detail on the benefits of using a minimal decoding unit and supervised training on intermediate coordinates.
3. Discuss the limitations of the Pallatom model in terms of its ability to generate proteins with diverse secondary structures. The results in Figure 2B and 2C show that Pallatom, like other comparative methods, has difficulty generating proteins with predominantly β-sheet structures. The authors should discuss the reasons for this limitation and potential strategies for addressing it in future work.
4.  **Explore the potential of Pallatom for designing larger and more complex protein systems, such as antibody complexes and self-assembling materials.** The authors mention this as a direction for future work, but they could expand on the specific challenges and opportunities involved in extending the model to these more complex systems.

---

> ### Author Response · Authors · 2024-11-25
> **Rebuttal by Authors**
>
> **W1:**
>
> Due to limited computational resources, we are unable to conduct detailed ablation studies during the rebuttal period to help readers better understand the model's interpretability. Based on our development experience, we offer the following insights for reference:
>
> 1. We continued using the single and pair representations from AlphaFold2 and AlphaFold3. We believe that high-quality pair representations significantly improve the quality of generated protein backbones, as supported by findings in Proteus [1].
>
> 2. To allow the gradient of the diffusion RMSE loss to directly influence the pair features, we proposed a key recycling mechanism that injects the generated 3D coordinates into the pair features through RBF. Unlike gradient-free recycling across the entire model in AF2 or AF3, our mechanism performs fast, iterative recycling within each unit while maintaining gradient propagation. We observed that this injection of pair bias effectively influences the updates of pair features.
>
> [1] Wang, Chentong, et al. "Proteus: exploring protein structure generation for enhanced designability and efficiency." bioRxiv (2024): 2024-02.
>
> **W3:**
>
> Thank you for your suggestion. At present, our work focuses on standard amino acids. However, we believe that Pallatom, benefiting from the existing input conformer feature channels, has the potential to extend to generating proteins capable of sensing ligands given their conformations. We plan to explore this direction in future work.
>
> **Q1:**
>
> Through our survey of related works, we identified certain limitations in existing methods for jointly designing sequences and structures, exemplified by the SOTA method Multiflow. Proteins are inherently multimodal, with complex and tight correlations between sequences and structures, forming a challenging joint distribution. Decomposing these into independent representations for separate sequence and structure diffusion disrupts this correlation, hindering the model's ability to learn valid sequence-structure combinations. For example, Multiflow exhibits poor matching between its designed sequences and structures (CO-DESIGN1: DES-aa=62.74%), significantly lower than the designability of sequences redesigned for backbones (PMPNN 1: DES-bb(w)=84.69%).
>
> To address this, we sought a representation that unifies sequences and structures in a single space. Full-atom protein structures inherently represent both sequence and structure. Thus, we developed the atom14 representation, which supplements virtual atoms to retain only pure coordinate information, enabling the representation of unknown amino acids in $\mathbb{R}^{14 \times 3}$.
>
> **Q2:**
>
> Thank you for the suggestion. The diffusion module in AlphaFold3, based on PairFormer, conditions on single and pair features and comprises three components: AtomAttentionEncoder, DiffusionTransformer, and AtomAttentionDecoder. However, in generative tasks, where no pre-encoded single or pair features exist, the existing framework is entirely unsuitable.
>
> Considering that diffusion RMSE loss primarily carries gradients in the coordinate dimension, we devised a recycling mechanism that rapidly feeds coordinate information back into the pair features. This mechanism is implemented within a minimal decoding unit as follows:
> - Update single features at the token level;
> - Broadcast token-level information to the atomic level for structural decoding;
> - Recycle decoded structural information back to token-level pair features.
>
> This unit design, combined with supervision on intermediate coordinates generated by each unit, enables the decoder layers to quickly learn decoding patterns, capturing finer distributions from backbones to side chains.
>
> **Q3:**
>
> Thank you for your suggestion. Based on our sampling results, we found that most methods struggle to generate predominantly β-sheet (EEE) structures. We attribute this to two main factors:
>
> 1. Within the selected protein length range, EEE structures exhibit lower structural diversity and sample frequency compared to other secondary structures. This data bias likely contributes to the difficulty of generating EEE structures.
>
> 2. Folding patterns for EEE structures are more challenging than those for predominantly alpha-helix (HEL) structures. β-sheets require longer primary sequences and involve more complex long-range interactions, making them harder to learn and stabilize. Additionally, EEE structures demand strict adherence to secondary structure rules throughout the protein length, where any random noise or local minima during sampling can obstruct their formation. Thus, EEE structures are inherently more challenging to sample.
>
> We believe that adjusting sampling probabilities during training based on the abundance of secondary structure types is a simple and feasible approach to alleviate this issue.

---

> > ### Comment · Reviewer_oX8v · 2024-12-02
> >
> > I'm satisfied with authors' response.

---

### Official Review · Reviewer_omne · 2024-11-03

**Soundness:** 2
**Presentation:** 2
**Contribution:** 2
**Rating:** 3
**Confidence:** 5

**Summary:**

The paper proposed to design protein by directly generating all atom coordinates. Experimental results show improvements on newly proposed metrics describing designablility, diversity and novelty.

**Strengths:**

Improved evaluation scores on proposed metrics.

**Weaknesses:**

While the concept shows promise, the network architecture is actually not novel. The core of the model appears to be heavily based on AlphaFold3, with adaptations made to handle cases without input sequences and MSAs. In several statements, the authors may have overstated the novelty of their approach (see questions).

The paper lacks sufficient evaluations to become useful. The main benchmark focuses on the model's ability in designing monomers with 60-120 residues, while other important scenarios such as motif-scaffolding, loop design etc (see details in RFdiffusion) are ignored. In fact, in the industry and research of protein design, de novo design is a worst setting with regard to real applicability such as drug discovery and enzyme engineering.

On the benchmark of PMPNN 1 (which is the most common setting), Pallatom only out-performs baseline methods on newly proposed metrics. This makes the comparison less convincing, especially given that the model is not specifically designed to improve diversity and novelty.

**Questions:**

1. P2 066 "we propose a new amino acid coordinates representation, atom14" The atom14 notation is used no later than AF2 and is not new at all.

2. P4 167-170 "We find that if residual connections ... repeatedly broadcast and inappropriately accumulated" This so-called challenge is not well-described. Readers cannot find any evidence in supporting the statement, which seems to be describing a bug in the author's residual network modeling.

3. All algorithms displayed in the appendix are almost directly copied from AF3's supplementary. The authors should either reduce the copied contents or justify why such massive copying is necessary.

---

> ### Author Response · Authors · 2024-11-25
> **Addressing Misconceptions About Innovation in Network Design**
>
> The generation of full-atom proteins faces a core challenge in representing unknown amino acid atoms, particularly in the absence of sequence information. Addressing these issues requires redesigning the model to solve a set of unique problems, fundamentally different from the protein structure prediction task exemplified by AlphaFold3. This core distinction leads to entirely different approaches in model design. Furthermore, in this field, reusing well-validated and efficient network modules is a common and reasonable practice. For instance, the network frameworks of FrameDiff and Multiflow largely retain the IPA module from AlphaFold2, which has not diminished their significant contributions to the protein design domain.
>
> Therefore, we argue that leveraging well-validated network components from AlphaFold3 as part of an efficient encoding strategy is not only reasonable but should not undermine the innovative nature of our work. Our method focuses on addressing fundamentally new challenges in full-atom protein generation, rather than merely adapting or modifying existing models.

---

> > ### Author Response · Authors · 2024-11-25
> > **Rebuttal by Authors**
> >
> > **W1:**
> >
> > This study introduces a novel full-atom protein design model and demonstrates its strong generative capabilities on small to medium-sized proteins. Compared to the state-of-the-art sequence-structure co-design method Multiflow, our approach achieves significant improvements in performance. Notably, the conference paper Multiflow also only conducted experiments on unconditional generation. In future work, we plan to extend our model to applications such as motif-scaffolding benchmarks and binder design, and to validate its practical utility through wet-lab experiments.
> >
> > **W2:**
> >
> > We believe that incorporating the pLDDT constraint into the designability metric (i.e., DES-bb(w)) is highly reasonable. As you mentioned, RFdiffusion adopts strict filtering standards before conducting wet-lab experiments in practical applications. For example, in the binder design task, RFdiffusion defines designability with constraints such as pAE<10, RMSE<1Å, and pLDDT>80. Therefore, it is natural for us to apply stricter standards, including the pLDDT>80 constraint, in the unconditional protein design benchmark to ensure sequence confidence. We argue that the differences between DES-bb(w) and DES-bb(wo) should be viewed as a significant finding. The performance drop across various methods after adding the pLDDT constraint reveals that inflated metrics might mislead evaluations of a model's real-world effectiveness.
> >
> > **Q1:**
> >
> > Full-atom protein design requires a representation of unknown amino acid types at the atomic level. While the atom14 representation in AlphaFold2 records atomic information for 20 standard amino acids, the `ATOM14` proposed in this study is fundamentally different and is specifically designed for representing unknown amino acids in generative tasks. To our knowledge, apart from the `ATOM73` representation proposed by Protpardelle, no other suitable representation currently exists for such tasks.
> >
> > **Q2:**
> >
> > In each AtomDecoder unit, information from the token track is broadcast to the atom track. If shuttle representation is not used, and both token and atom information are updated through residual accumulation, broadcasting token information to the atom track will result in unnecessary duplication within the atom track.
> >
> > **Q3:**
> >
> > As described in the appendix, to avoid unnecessary repetition, we highlighted algorithms consistent with AlphaFold3 in blue. The pseudocode presented is entirely novel and specifically tailored for full-atom protein generation tasks.

---

### Official Review · Reviewer_erk6 · 2024-11-03

**Soundness:** 1
**Presentation:** 1
**Contribution:** 2
**Rating:** 3
**Confidence:** 4

**Summary:**

This study proposed a new architecture called Pallatom to generate protein structures. When the side chain coordinates are unknown, they are generated by a strategy named Atom14. The generative performance is evaluated and compared with several existing methods in terms of designability, diversity, and confidence.

**Strengths:**

1. The paper provides a comprehensive description of the implementation of the proposed method.
2. The proteins generated by the method exhibit notably high levels of diversity and novelty, indicating considerably strong performance in these areas.

**Weaknesses:**

1. The rationale behind the study lacks clear explanation and supporting evidence, such as empirical justification for the importance of conducting all-atom protein generation.

2. The `atom14` representation seems rather trivial, as it mainly consists of initializing 14 atom entities per AA and padding extra atoms to the C_alpha. This method does not appear to represent a substantial novelty.

3. The methodology section is poorly articulated, making it difficult to assess the significance of the proposed modules or workflows in relation to existing frameworks.

4. The figures are challenging to decipher due to the small font size used.

5.  The evaluation criteria are not well-justified. For instance, why it is important for a model to design novel structures?

6. The overall presentation has a large room for improvement, with many typos to be fixed.

**Questions:**

1. In the statement "(A) series of protein generation models based on SE(3)" could you specify why it is referred to as recent when the references are from 2020 and 2021? Are there more recent studies that support this claim?

2. Could you provide evidence or references to clarify the challenges faced by "The P(structure | seq) · P(seq) strategy faces challenges when sampling in the high-dimensional (space)"? Also, could you explain why explicit side-chain interactions are considered essential in the context of the "P(backbone) · P(seq | backbone) strategy"?

3. Where does the statement "The ultimate goal of protein generation is to directly obtain a sequence along with its corresponding structure" originate from? Could you provide references and clear explanations to support this claim?

4. How was the number 14 determined for the `atom14` representation? How does this approach handle amino acids with more than 14 atoms, such as Tryptophan?

5. Could you clarify why the existing designability evaluation metric is not considered suitable for evaluating all-atom generation results, given that RMSD is calculated on atoms rather than C_alpha?

6. Why are there references to previous studies that used designability, but none for diversity and novelty? Could you provide references that discuss these aspects?

---

> ### Author Response · Authors · 2024-11-25
> **Rebuttal by Authors**
>
> **W1:**
>
> Side chains are critical carriers of functionality in protein active regions. For example, in enzyme active sites and protein-protein interaction interfaces, side chains play an essential role. Currently, most structural design methods in the field are limited to protein backbone design, with no incorporation of side-chain information in the models. Against this backdrop, designing a full-atom protein model that explicitly incorporates side-chain construction holds immense application potential and academic significance.
>
> **W2:**
>
> Full-atom protein design requires a comprehensive representation of unknown amino acid types. To the best of our knowledge, apart from the ATOM73 representation proposed by Protpardelle, no other representation currently exists for this purpose. Our proposed ATOM14 representation is a novel approach tailored for full-atom protein design. It is not intended as a record of amino acid atomic information, as seen in AlphaFold2, but rather as a unified description of unknown amino acids in generative tasks.
>
> **W5:**
>
> We believe that several highly influential studies in the field, such as RFdiffusion and its applications [1, 2, 3], are all advancing the direction of de novo protein design. De novo protein design aims to create entirely new proteins with specific structures and functions from scratch, rather than modifying existing natural proteins. Evaluating generated proteins requires multiple metrics. On the basis of designable proteins, it is natural to pursue higher diversity to prevent the model from collapsing into generating only a limited set of structures. At the same time, greater novelty is necessary to ensure that the generated proteins are not merely imitations of natural ones. If the generated proteins are too similar to existing natural proteins, this would contradict the primary goal of de novo design: exploring protein functionalities that do not yet exist in nature.
>
> [1] Watson, Joseph L., et al. "De novo design of protein structure and function with RFdiffusion." Nature 620.7976 (2023): 1089-1100.
>
> [2] Bennett, Nathaniel R., et al. "Atomically accurate de novo design of single-domain antibodies." bioRxiv (2024).
>
> [3] Liu, Yufeng, et al. "De novo protein design with a denoising diffusion network independent of pretrained structure prediction models." Nature Methods (2024): 1-10.

---

> > ### Author Response · Authors · 2024-11-25
> > **Rebuttal by Authors**
> >
> > **Q1:**
> >
> > These two references pertain to existing studies on SE(3) invariance or equivariance networks. Descriptions and citations related to generation models can be found in the Related Work section.
> >
> > **Q2:**
> >
> > **Challenges of the P(structure | seq) · P(seq) strategy in high-dimensional sampling:**
> >
> > 1. Complexity of the energy landscape:
> >
> > The P(structure | seq) · P(seq) strategy heavily relies on the energy landscape of structure prediction models [1]. Exploring the gradients of P(structure | seq) is inherently challenging. For example, the latest representative work RSO [1] highlights the necessity of using the softseq representation instead of onehot sequences, illustrating the difficulty of direct gradient computation. Moreover, the gradient updates in such models rely on empirical parameter adjustments, which underscores the ruggedness and complexity of the P(structure | seq) space. The authors of RSO also emphasize the importance of randomly initializing sequences and optimizing structure and sequence simultaneously in local minima.
> >
> > 2. Low sample efficiency:
> >
> > Sampling methods in high-dimensional spaces are inefficient, leading to greater computational costs and increased time consumption.
> >
> > [1] Frank, Christopher, et al. "Scalable protein design using optimization in a relaxed sequence space." Science 386.6720 (2024): 439-445.
> >
> > **Importance of side-chain interactions in the P(backbone) · P(seq | backbone) strategy:**
> >
> > This strategy first generates a protein backbone structure P(backbone), followed by sequence prediction conditioned on the backbone, P(seq | backbone). However, a critical limitation is the lack of side-chain information during backbone generation, which may fail to capture intricate interactions, particularly in functional regions with loopy structures. The inability to account for side-chain influences significantly constrains this approach in designing complex functional proteins, presenting considerable room for improvement.
> >
> > **Q3:**
> >
> > De novo protein design refers to creating entirely new proteins with specific structures and functions from scratch. The ideal way to document such proteins is to provide the corresponding sequences that can be expressed and folded into the designed structures. Our Pallatom method achieves this by directly generating the full-atom structure of the target protein, simultaneously providing both a coherent sequence and structure.
> >
> > **Q4:**
> >
> > Currently, we focus on the heavy atoms of 20 standard amino acids, following the recording conventions of AlphaFold2. For TRP (tryptophan), the 14 recorded atoms are:
> >
> > [N, CA, C, O, CB, CG, CD1, CD2, NE1, CE2, CE3, CZ2, CZ3, CH2]
> >
> > **Q5:**
> >
> > When the model is capable of generating full-atom proteins, the quality of side-chain atoms can also be evaluated. Compared to Cα RMSD, all-atom RMSD is a stricter metric as it accounts for the positional accuracy of side-chain atoms.
> >
> > **Q6:**
> >
> > The metrics for diversity and novelty are also adopted from Multiflow, continuing its approach to evaluating generative quality.

---

### Official Review · Reviewer_aE3n · 2024-11-04

**Soundness:** 2
**Presentation:** 1
**Contribution:** 2
**Rating:** 3
**Confidence:** 4

**Summary:**

The authors propose Pallatom, a model for all-atom protein generation. The authors design atom14 to represent the unknown amino acid side-chain coordinates.

**Strengths:**

1. The authors define three kinds of metrics, DES, SIV and NOV, and show Pallatom has excellent performance on these metrics.
2. Pallatom considers the information from side-chains.

**Weaknesses:**

1. There is no formal definition for the defined metrics, and the authors only claim that these metrics are evaluated by TM-score, which makes it difficult to follow the authors' idea. I'm not sure if DIV can be an ideal metric, since the final target is to generate protein structures that are similar to natural proteins. High diversity indeed indicates the design ability of Pallatom, but the authors should also investigate whether high diversity aligns with natural proteins.

2. The paper is difficult to follow. When the authors refer the appendix in the main body, please tell the readers where we can find exactly in the appendix. Please do not just let us find the content in appendix by ourselves, given that you have 12 pages in appendix.

3. The application seems limited, because only the unconditional experiments are conducted.

**Questions:**

I am really surprised to see that in the training data, the authors only take approximately 7k and 27k structures from PDB and AFDB, respectively. Are these structures sufficient for training this model? Additionally, I find that the authors set the maximum training length as 128. What is the motivation behind this decision?

---

> ### Author Response · Authors · 2024-11-25
> **Rebuttal by Authors**
>
> **W1: Rationality of Designability, Diversity, and Novelty as Metrics**
>
> Designability, diversity, and novelty are three widely adopted core metrics in the field of protein structure design. Similar to MultiFlow, this study builds upon the metric framework established by the seminal method FrameDiff. In response to the reviewer’s concerns, we provide a more detailed explanation below:
>
> **1. Regarding “I'm not sure if DIV can be an ideal metric, since the final target is to generate protein structures that are similar to natural proteins”**
>
> We believe that highly influential works in the field, such as RFdiffusion and related studies [1, 2, 3], have been striving toward de novo protein design. De novo protein design refers to creating entirely new proteins with specific structures and functions from scratch, rather than making local modifications to existing natural proteins. Therefore, the goal of protein design should not be limited to generating structures similar to natural proteins. Instead, evaluating generated proteins requires multiple metrics: on the basis of designable generated proteins, it is natural to pursue higher diversity, ensuring that the model does not collapse into generating only a limited number of structures; and better novelty, ensuring that the generated proteins do not overly imitate natural proteins.
>
> [1] Watson, Joseph L., et al. "De novo design of protein structure and function with RFdiffusion." Nature 620.7976 (2023): 1089-1100.
>
> [2] Bennett, Nathaniel R., et al. "Atomically accurate de novo design of single-domain antibodies." bioRxiv (2024).
>
> [3] Liu, Yufeng, et al. "De novo protein design with a denoising diffusion network independent of pretrained structure prediction models." Nature Methods (2024): 1-10.
>
> **2. Regarding “but the authors should also investigate whether high diversity aligns with natural proteins”**
>
> Overemphasizing similarity to natural proteins risks leading to “overfitting,” where the model only makes minor modifications to existing proteins. This approach limits the potential of de novo design, hindering the discovery of novel functionalities or proteins tailored to specific needs (e.g., drug development or target-specific proteins). Such a strategy contradicts the primary goal of de novo design: exploring functionalities absent in nature. To address this, we provide Appendix Figure 7, which showcases numerous highly novel designable proteins, highlighting Pallatom's robust capability for de novo protein design.
>
> **W2:**
>
> We acknowledge that the current draft includes excessive references to existing concepts, which may impose a cognitive burden on readers. However, from a technical perspective, implementing a novel protein design paradigm requires addressing numerous intricate details that cannot be fully elaborated in the main text. To mitigate this, we have provided pseudocode for each innovative module in the appendix, aiming to clearly present the core computational processes. Additionally, the appendix includes detailed descriptions of the dataset, model input features, and model details, enabling readers to gain a comprehensive understanding of our work.
>
> **W3:**
>
> This study proposes a novel full-atom protein design model and demonstrates its robust generative capabilities on small to medium-sized proteins through extensive validation. Compared to the current state-of-the-art sequence-structure co-design method, Multiflow, our approach achieves significant performance improvements. Notably, the conference paper Multiflow also focuses exclusively on unconditional generation experiments. In future work, we plan to extend our model to applications such as motif-scaffolding benchmarks and binder design, while conducting wet-lab experiments to validate its practical utility.
>
> **Q1:**
>
> We trained the model solely on structural data, which is sufficient to endow it with strong generative capabilities. The rationale for setting the maximum training length to 128, as described in the main text, aligns with industrial applications, where small proteins can be easily synthesized using commercial oligo-pool methods. We will further expand the training scale to address more complex computational scenarios.

---

### Official Review · Reviewer_7q6Q · 2024-11-06

**Soundness:** 3
**Presentation:** 1
**Contribution:** 3
**Rating:** 5
**Confidence:** 2

**Summary:**

The paper presents Pallatom, a generative model for protein structures based on all-atom coordinates. Precisely, the standard all-atom representation is extended to obtain the $\texttt{atom14}$ representation, which pads the original 6 all-atom features with 8 more virtual atoms. At its core, Pallatom is a diffusion generative model which is trained to denoise the $\texttt{atom14}$ representations, adapting parts of the AlphaFold3 model from the structure prediction to the generative setting. Experiments in _de novo_ protein design show that Pallatom is capable of generating coherent and diverse low-to-medium-sized proteins.

**Judgment**

In my opinion, this paper is a tremendous effort in solving the de novo protein generation task with denoising diffusion. However, I could not fully appreciate the author's contribution because the way it is presented is rather convoluted and because of some unfortunate choices (such as the rather confusing notation borrowed from another paper). I am sure this has to do with the fact that the authors had to limit the writing to 10 pages. Perhaps ICLR is not the right venue for this kind of papers, because of the page limits; probably a journal article would do more justice to such a technically involved work. I am giving a score of 5 for the moment being, I'll be happy to revise after hearing from my fellow reviewers and the author's replies.

**Strengths:**

- the paper tackles a very difficult task, with a potentially disrupting impact in the comp-bio area.
- the overall design appears very thoughtful. Adapting AlphaFold3 designs to the generative process is a real challenge, which judging by the experimental results has been mastered.
- the $\texttt{atom14}$ representation is innovative and seems effective, although its importance has not been verified by ablations.
- experimental results are excellent, although they are measured on custom metrics.

**Weaknesses:**

**Disclaimer**

I couldn't properly understand the methodological contribution of this paper (details below). This might bias my overall judgment.


**Weaknesses**

The main problem with this paper is the way it is presented. While I had no problems in understanding papers on the same subject (e.g. Protpardelle), here I had a very difficult time reading. Some reasons why:

1) the paper takes too many concepts from granted (e.g., the all-atom representation is not explained, examples of sequence/structure conflicts that motivate the $\texttt{atom14}$ representation are lacking, and so on). Perhaps it would be better to add a brief explanation of the all-atom representation in the introduction, and providing 1-2 specific examples of sequence/structure conflicts in Section 2.1 to motivate the $\texttt{atom14}$ representation.

2) it uses the unfortunate notation of Karras et al. (2022), which makes understanding the contribution more difficult (e.g. $\sigma(t)$ vs. $\sigma_t$, $F_{\theta}$ undefined). Perhaps you could add a notation table to help the reader. Also please make sure that all the symbols and acronyms used in the paper are defined before or immediately after their first use.

3) it is not well-structured, or at least, it uses section names that are confusing (e.g. section 2 is about "preliminaries", yet some contributions, e.g. $\texttt{atom14}$ are described there). Maybe it would be better to reorganize the sections so that your contribution stands out from work that has already been done, although I understand this could be a very disruptive change.

Other major problems I have found:
- $\texttt{atom14}$ is not ablated, it is not clear whether its addition is really needed to improve performances. It would be better to include a specific ablation on Pallatom with and without the $\texttt{atom14}$ representation, using the metrics you defined for all-atom models.
- comparing backbone-only models with their own metrics to all-atom models with their own metrics is not really meaningful in my opinion. However, I'm not sure there is a metric that could be used in this case; perhaps you can just compare the ratio of novel and unique protein structures produced by both classes of methods?

**Questions:**

- why representations are initialized with the standard conformation of alanine and not other amino acids (e.g. methionine)?
- Pallatom is evaluated with custom metrics. Can you provide the scores obtained by Pallatom in the metrics used by competitors (e.g. Protpardelle)?

**Details Of Ethics Concerns:**

None.

---

> ### Author Response · Authors · 2024-11-25
> **Rebuttal by Authors**
>
> Thank you for recognizing our work and encouraging its future prospects. Indeed, we are committed to surpassing the conventional SE(3)-diffusion-based backbone design paradigm in protein design and pursuing a novel framework capable of end-to-end full-atom protein design. We acknowledge that the current manuscript contains an excessive reliance on assumed knowledge, which might hinder readers' understanding. Unfortunately, due to space constraints, we were unable to elaborate on these background concepts in detail. Additionally, we recognize the issues in the manuscript's organization and presentation. We will carefully incorporate your suggestions to improve our work in future iterations. Thank you for raising these questions. Below are our detailed responses to each issue:
>
> **Major Problem 1: `ATOM14` is not ablated**
>
>  This issue likely arises from our insufficient explanation of the `ATOM14` representation proposed for unknown amino acids in generative tasks. In protein structure prediction tasks (e.g., AlphaFold3), the number and order of atoms in 20 standard amino acids are determined by the sequence. For instance, given a 3-mer segment:
> - Token track: 'CAS'
> - Atom track: '(N, CA, C, O, CB, SG), (N, CA, C, O, CB), (N, CA, C, O, CB, OG)'
>
> In generative tasks, however, the sequence is unknown. To avoid amino acid type leakage due to differences in atomic counts, we homogenize each amino acid in the coordinate space. We propose `ATOM14`, where 14 coordinates are sampled independently from a 3D Gaussian noise distribution to represent an unknown amino acid. From the model’s perspective, the initial sampled coordinates (14×3) for each amino acid are indistinguishable Gaussian distributions.
> Since real amino acids do not all have 14 atoms, we pad the number of atoms to 14 for each amino acid and overlap the extra virtual atoms with the CA atom, which represents the amino acid's center.
> Therefore it is the foundational input format for Pallatom to perform full-atom protein design. Ablating `ATOM14` would make it impossible for Pallatom to design full-atom proteins from Gaussian noise. While other representations exist (e.g., Protpardelle’s `ATOM73`), they demand significantly more memory, and the metrics of Protpardelle in Table 1 indicate that `ATOM73` may not be an optimal choice.
>
> **Major Problem 2: Metric Refinements**
>
>  Protein backbone design is typically evaluated on three metrics: designability, diversity, and novelty. In PMPNN 1 mode, we adopt the same three metrics as in Multiflow, adding a pLDDT > 80 filter to the designability metric. This is a natural addition, as lower pLDDT scores indicate lower confidence.
> In the full-atom protein generation scenario (CO-DESIGN 1 mode), we extend the designability metric from Cα RMSD < 2 to all-atom RMSD (aa-RMSD) < 2, as full-atom design requires considering the distribution of side-chain atomic coordinates. Notably, aa-RMSD is a stricter metric; aa-RMSD < 2 implies Cα RMSD < 2.
>
> **Question 1:**
>
>  The 20 standard amino acids have distinct side-chain conformations, but their backbone atoms (N, CA, C, O) exhibit nearly identical conformations. Starting from unknown amino acids avoids side-chain information leakage. Given that all amino acids except GLY have a CB atom, we use the backbone conformation of ALA (including CB), which is generalizable to most amino acids and helps the model quickly learn the local distribution of backbone atoms.
>
> **Question 2:**
>
>  For designability, Protpardelle’s scRMSD is essentially identical to DES-bb (without pLDDT) in PMPNN 1 mode, and scTM evaluates designability from a TM-score perspective. For diversity, Protpardelle modifies the structure cluster count/total designable structures ratio we use, but the core idea is similar. In CO-DESIGN 1 mode, we additionally evaluate sequence diversity (DIV-seq). For novelty, Protpardelle assesses the nearest neighbor TM-score (nnTM) between generated structures and the training set. Similarly, Pallatom evaluates the nnTM between generated structures and natural proteins in the PDB database. Since the training sets of all methods include the PDB database, these evaluation approaches are highly consistent.

---

### Note · Authors · 2025-01-21

I have read and agree with the venue's withdrawal policy on behalf of myself and my co-authors.